# Towards a Recommender System for In-Vehicle Antenna Placement in Harsh Propagation Environments

**DOI:** 10.3390/s22176339

**Published:** 2022-08-23

**Authors:** Daniel Kraus, Konrad Diwold, Jesús Pestana, Peter Priller, Erich Leitgeb

**Affiliations:** 1Pro2Future GmbH, 8010 Graz, Austria; 2Institute of Technical Informatics, Graz University of Technology, 8010 Graz, Austria; 3AVL List GmbH, 8020 Graz, Austria; 4Institute of Microwave and Photonic Engineering, Graz University of Technology, 8010 Graz, Austria

**Keywords:** Bluetooth Low Energy (BLE), antennas, wireless sensor network, recommendation system, ray tracing

## Abstract

This paper presents a novel approach to improving wireless communications in harsh propagation environments to achieve higher overall reliability and durability of wireless battery powered sensor systems in the context of in-vehicle communication. The goal is to investigate the physical layer and establish an antenna recommendation system for a specific harsh environment, i.e., an engine compartment of a vehicle. We propose the usage of electromagnetic (EM) and ray tracing simulations as a computationally cost-effective method to establish such a recommendation system, which we test by means of an experimental testbed—or test environment—that consists of both a physical, as well as its identical simulation, model. A pool of antennas is evaluated to identify and verify antenna behavior and properties at specified positions in the harsh environment. We use a vector network analyzer (VNA) for accurate measurements and a received signal strength indicator (RSSI) for a first estimation of system performance. Our analysis of the experimental measurements and its EM simulation counterparts shows that both types of data lead to equivalent antenna recommendations at each of the defined positions and experimental conditions. This evaluation and verification process by measurements on an experimental testbed is important to validate the antenna recommendation process. Our results indicate that—with properly characterized antennas—such measurements can be substituted with EM simulations on an accurate EM model, which can contribute to dramatically speeding up the antenna positioning and selection process.

## 1. Introduction

In the evolving modern vehicle design, development and manufacturing environments, there exists a trend towards ubiquitous computing, where the number of sensing units and their communication interfaces are rapidly increasing. In vehicles, this trend increases the size and weight of the in-vehicle network and its wiring harness, which complicates its installation and leads to many issues [1]. Wireless communications can produce relief in this context. For a new project involving numerous wireless sensor systems, the wireless communication setup is complex and optimizing it can be costly and demanding [2].

Harsh environments (or harsh propagation environments) are challenging for any kind of wireless communication [3]. A link is heavily influenced by the materials of the environment as well as by other wireless signals [4,5,6]. Thus, a wireless link might fail due to physical effects—multipath propagation being a prominent example [7,8].

Using the 2.4 GHz wireless communication band—where many technologies such as WiFi, Bluetooth, Zigbee are utilized—aggravates attempts to establish a reliable wireless network structure [9]. In this work, we focus on Bluetooth Low Energy (BLE), since BLE can be applied for basically any wireless short-range application and constitutes the most attractive low-cost solution with COTS devices due to its low energy consumption. In our previous work, the robustness and applicability of BLE with COTS devices was explored [10,11].

Most approaches to improving and optimizing wireless communications in harsh environments tackle protocols and network architecture [12], but not the underlying physical attributes. Often, an adaptive structure is established which improves the data throughput by using alternative paths from transmitter to receiver [13,14,15,16,17]. In contrast, this work considers commercial off-the-shelf (COTS) antennas and devices and the automatic enhancement of sensor nodes by equipping them with an antenna from a pool of available COTS antenna types, which constitutes the motivation for our developed antenna recommender system.

The recommender system was developed in cooperation with AVL List GmbH. In the context of the vehicle manufacturing industry, the goal was to establish a reliable and durable wireless network within testing benches and test stands as well as prototype vehicles. The problem in such evaluation environments is that the effort to provide wiring to each sensor location is immense and often prone to errors. Instead, when achieving the desired goal by means of a functional wireless sensor network, the effort to setup the communication network is considerably reduced and potential fault sources during installation of the wiring are removed. Previous investigations at the AVL test stands aimed at characterizing the harsh electromagnetic (EM) channel in an engine testbed when investigating different methods for the localization of passive ultra-high frequency (UHF) radio frequency identification (RFID) tags [18,19].

In this work, the conducted investigations were aimed at a temporary instrumentation of vehicles with additional wireless sensor nodes (WSNs). BLE is the only technology that can satisfy the requirements of AVL in this context, as data rate (certain sensors require a bandwidth of at least 1 Mbps) and battery life both have to be sufficient to support the AVL use case. With the presented recommender system, test engineers obtained antenna recommendations for each wireless sensor location in the engine compartment, and by applying the recommended antennas, they improved the throughput of each individual node in the communications network.

An overview of this work is presented in Section 2.

### 1.1. State of the Art

Establishing wireless sensor networks in vehicles has been a focus of research in recent years. In this section, we assess whether and how the achievements of previous work have an impact on our presented approach. In Table 1, related research to the presented work is shown and measurements and experiments of similar approaches are compared based on their key parameters, i.e., the used protocol and frequency range, the type of the device, the data/sample rate with which the wireless communication was measured, the investigated range, and whether metal was near the transmitting devices during the investigations.

An approach to improve wireless communications in harsh environments is to utilize simulation models in order to estimate the behavior of communication nodes. In the early 2000s, wireless tire-pressure monitoring systems (TPMS) were integrated in modern vehicular networks. The work by Song et al. [20] studied wireless TPMS communication by means of a simplified three-dimensional electromagnetic (EM) simulation model, which consisted of a mesh (surface) model without obstacles. A similar approach was adopted by Zeng et al. [21], where the effect of the vehicle body on EM propagation in TPMS was investigated. Lasser et al. [22] extended the EM simulation approach and conducted channel measurements in the 866 MHz and 2.45 GHz band with two simple dual-band antennas for passive RFID tags.

Ray tracing is a very powerful simulation modeling technique for the propagation of wireless signals. In general, ray tracing is more commonly used for electrically large environments for tracing radio signals, e.g., in the ionosphere or on large objects such as airplanes [23,24]. Computing models of the size of a vehicle with conventional EM simulation models would take an excessively long time, as has been the case in previous investigations in similar environments [25]. In contrast, ray-tracing simulation is a simplification of the very complex EM simulation problem. The propagating rays are calculated by solving Maxwell’s equations with numerical methods [26]. If an obstacle is hit, the wave (=ray) will either be refracted, absorbed, reflected, or scattered. Depending on the size of the propagation environment, the calculations can be highly complex and computation intensive. Due to the computational effort, ray-tracing methods in the past seemed inadequate for more complex scenarios. However, in recent years, the computational power, availability, and affordability reached a level where it makes sense to continue with the investigations. Since models are constantly evolving and becoming more detailed, ray tracing is by now even applied for small-scale scenarios, as is the case with on-chip wireless communications [27].

Another approach to improve wireless communications is to modify the communication protocol itself. Two examples of such an approach are in [28], where the coexistence of Zigbee and Bluetooth is examined, and in [29], where Zigbee and Bluetooth protocols were investigated. In both cases, the system performance was then evaluated in terms of packet error rate (PER) and average/peak packet latency.

Yet another approach is to study the influence of the immediate communication environment on the wireless communication—such as metallic cages, obstacles, and people—by means of various types of experimental measurements related to wireless link quality. RSSI and packet reception rates (PRRs) have been frequently used to determine the applicability of wireless sensor nodes for in-vehicle communications, as shown in [13,14,16,30]. In the investigations of [16], the path loss and path loss exponent values could be deduced from the received signal strength measurements in a vehicular environment. Similar measurements were performed in [15], but the focus there was more on the measurement of the channel and the identification of potential sources of electromagnetic interference (EMI). Another approach for characterizing the in-vehicle wireless communication channel was introduced by Herbert et al. [31] to understand both the time variation and spectral properties.

Regarding interference through obstacles, in [32] the authors measured the power spectrum density (PSD) in in-vehicle scenarios with multi-sensor—or mesh—networks to study the interferences caused by passengers and other objects.

Other research in this context is focused on channel estimation and beamforming of smart antennas [33,34], where algorithms such as the leaky least mean square (LLMS) algorithm are applied to mitigate intersymbol interference in high-data-rate transmissions [35]. These approaches can be realized in most cases with antenna arrays. With beamforming of smart and adaptive antennas, it is possible to reduce side lobes and direct the main lobe towards the transmitting/receiving antenna, so that the data throughput and the reliability of the wireless transmission system are enhanced. In this work, we are focused on COTS antennas, and will not investigate adaptive antenna systems.

### 1.2. Research Contribution

Our research focuses on engine compartments, which constitute an enclosed Faraday cage—physical processes, such as EM reflection and scattering, are ubiquitous in these harsh environments. With this aim in mind and in accordance with our industry partner, we selected the components of our experimental testbed to establish a physical test setup that displays EM characteristics similar to those of a real engine compartment. The experimental testbed enabled us to perform our research on EM and ray-tracing simulations and achieve a prototype antenna recommender system for specified positions in harsh propagation environments.

Other researchers have used EM simulations to design an antenna array to achieve the localization of UHF RFID tags inside harsh propagation environments [18,19]. In contrast, we investigate the utilization of EM and ray-tracing simulations to achieve antenna recommendations for specific antenna positions and orientations in harsh propagation environments. We also test it in our experimental testbed and analyze whether the utilization of experimental measurements produce the same antenna recommendations as their corresponding EM and ray-tracing simulation results. In addition, we test our prototype implementation of the recommender system using COTS BLE dongles and antennas, which—among others—pose further challenges in the creation of accurate EM simulation models of the antennas. These analyses and the implementation of the prototype antenna recommender system for antenna placement in harsh propagation environments constitute our first research contribution.

Our second contribution is the investigation of the possibility—in the context of the COTS antenna recommendation problem—to substitute the requirement of countless antenna and channel measurements with cost-effective electromagnetic (EM) and ray-tracing simulations. There have been investigations in the past where ray tracing was applied to simplified engine compartment models to estimate wave propagation [36]. In comparison to this work, we use EM and ray-tracing simulations to predict the effect of the positioning and orientation of the antennas on the wireless communication, as well as of the presence of obstacles. In comparison to works studying the influence of the propagation environment and obstacles [15,16,32], our focus is specifically on understanding the usability of EM and ray-tracing simulations to substitute experimental measurement efforts.

## 2. Concept Overview

Figure 1 gives an overview of the article. In order to provide antenna recommendations, we created accurate EM and ray-tracing models and simulations for our experimental testbed, which behaves electromagnetically very closely to an engine compartment. The full EM model consists of: the EM characterization of the receiver (RX) COTS BLE device and its internal antenna (Section 5), the EM model of the transmitter (TX) COTS BLE antenna (Section 6.6), and the EM toolbox and obstacle models (Section 4). The reasoning behind the selection of the metallic toolbox, which substitutes the engine compartment in our experiments, is explained in Section 3.

The full EM model was validated through measurements on the experimental testbed and tested with several COTS BLE antennas (Section 6.1). For this purpose, we utilized both, a vector network analyzer (Section 5.2.2), to achieve very accurate measurements of antennas and channels, and the RSSI values (Section 5.2.1), which are self-reported by the COTS BLE dongles or boards.

A wireless communication scenario is defined by the positions inside the metallic toolbox of the antennas of the TX and RX devices, and that of the obstacle. For each scenario, the scattering parameters (S-parameters) can be derived either from the VNA measurements or from the EM simulation results. The S parameters of the scenario—either acquired or simulated—form the basis to validate and benchmark the accuracy of the EM simulations (Section 6 and Section 6.4).

The inputs to the antenna recommender system (Section 6.8) are the setup scenario along with the selection of antenna candidates (Section 6.1)that are to be tested—either experimentally or by means of EM simulations—each with its EM antenna model. The wireless communication results for all tested antennas are gathered in a database and utilized to calculate the antenna ranking for each scenario by means of the S parameters, either acquired by measurement or simulation. The comparison of the measurement against the simulation-based antenna rankings is used to benchmark the recommender system.

## 3. Experimental Testbed—Physical Mock-Up of an Engine Compartment

A well-defined experimental testbed—for which an accurate EM simulation model can be created—has many advantages over a real engine compartment. The testbed allows the adaptation of the test scenarios to specific needs and to change the EM environment sequentially—both in simulation and experimentally—which ultimately eases the identification of areas and aspects of interest. In addition, a car engine compartment cannot just be tested and measured in a laboratory without enormous effort.

If the wireless system were directly installed inside a real engine compartment, such experiments would be costly and time-consuming. Instead, by means of our experimental testbed, we can investigate the COTS BLE antenna evaluation and placement problem in an efficient manner.

Therefore, a major focus for our experimental testbed is to display electromagnetic characteristics similar to those of a real engine compartment, allowing the verification through measurements of the designed antenna recommender system, the EM and ray-tracing simulations, and their predictions about the performance of the COTS BLE antennas. The technical characteristics of the testbed have been aligned with our company partner AVL to achieve a highly accurate model for verification.

Our experimental testbed (Figure 2 and Section 6.1) consists of: a physical mock-up of an engine compartment—a metallic toolbox—a movable metallic object, and a pair of wireless sensor nodes (WSNs) used for wireless communications. The dimensions of the metallic toolbox are L = 1500 × W = 500 × H = 500 (mm). The testbed is completed with accurate EM models (Section 4) of the physical setup, which, for the toolbox, is achieved by means of the detailed CAD model and material properties of the toolbox.

The toolbox weighs around 13 kg—i.e., it is transportable—and it was the most affordable option with material properties close to those of an engine compartment. A common vehicle bodyshell is made of aluminium sheets of alloys 5182, 5754 or 6111 with a material thickness in the range of 0.8 mm to 1.8 mm [37]. In comparison, the toolbox has a maximum panel thickness of 3 mm and is made of aluminium alloy 5754 (EN AW-5754). The electrical conductivity of this alloy 5754 at room temperature (20 °C) can be assumed with 18 m/(Ω-mm^2^) [38].

## 4. Electromagnetic Simulations of the Experimental Testbed

For the EM-simulation-based antenna recommender system to work, it is required to create an accurate electromagnetic (EM) model that represents the real wave propagation environment with enough precision for the EM simulation.

To achieve precise EM simulations in RF harsh propagation environments, an accurate EM model of a physical test environment has to be created. The model is characterized by the geometry of the target environment, including all objects and material properties [39].

Moreover, the model must include the layout of static objects and surfaces, as well as of all moving objects within the environment.

Separately, all objects inside the environment that emit EM energy or radio signals must be mapped into the model because these emissions can interfere with wireless communications. This means that an evaluation and performance analysis of a wireless device in a specific environment at a defined position is only valid for exactly that setup scenario.

This motivated the design of our physical testbed (Section 3) along with the creation of its full EM model (Section 4) to perform experiments and simulations of each setup scenario.

### 4.1. Full Electromagnetic Model

Our experiments are executed on the experimental testbed shown in Figure 2 and Section 6.1. One of the reasons to select this metallic box was that it is very well-defined, as a detailed CAD model of the toolbox is available and the material properties of the metal encasings are known.

We used CST Studio Suite 2021 for the electromagnetic evaluations and simulations—including ray tracing, see Section 4.2. The EM model of the toolbox was created in CST with its surfaces defined as sheets. As the EM model represents the closed toolbox, the EM simulations will only be accurate for experiment scenarios where the testbed’s lid is shut—or closed. The cable connection to the devices is not considered in the EM modeling, because for the VNA measurements, the endpoints are calibrated to remove these influences in the measurements.

The full EM model for each of our experiments consists of three parts: the environment—i.e., the toolbox’s and the obstacle’s EM models—the transmitter antenna and the receiver antenna. This means that we had to create accurate EM simulation models for each of the tested COTS antennas, which requires iterative testing (Section 6.6).

Therefore, for the creation of the full EM model, the component EM models were imported into the system assembly module of CST Studio Suite. A simulation project then had to be created with a hybrid solver task for bi-directional communication—through experimentation we identified the asymptotic solver with applied ray tracing as the best-fitting option for our work. In Table 2, the required changes for the asymptotic solver setup of the EM environment are listed—that is, all other settings remain unchanged.

In our experience, the listed settings provided accurate results in terms of the prediction of the S parameters, which we have compared against their VNA-derived counterparts in numerous experiments (see Section 6.4). Therefore, we believe that—regardless of the environment and the antennas—the listed settings can be applied to similar setups, e.g., to simulate wireless transmission in frequency bands similar that used by BLE.

### 4.2. EM Ray-Tracing Simulation

As a post-processing step on the EM simulation results and related meta-data, the asymptotic solver uses the so-called shooting-bouncing ray (SBR) method to approximate the propagation of radio waves from antennas. In this approach, rays are randomly launched in all directions from a defined transmitter. Then, these rays are traced to the point where a defined stopping criteria is attained—e.g., either hitting the receiver or the maximum ray length or number of ray reflections are reached. At the receiver location, a box or a sphere is defined as a target volume for the launched rays to hit. After all rays have been simulated, all intersecting rays are displayed.

The complexity of the calculations is dependent on the complexity of the EM environment and the number of launched rays. That is, the computation time increases linearly with the number of launched rays [40]. Applying ray-tracing simulation to the model in Figure 3 requires sufficient computational power (e.g., a computing cluster) to obtain the results in a reasonable time. In contrast, our experimental testbed represents a simpler EM wave propagation environment (see Figure 4 and Section 6.7). In our case, we were limited to an AMD Ryzen Threadripper 2970WX with 128 GB RAM and a basic licence of CST Studio Suite 2021. If the final number of frequencies sampled in the simulations is limited, results are attained in a reasonable time using our computing hardware (HW), which has—through this work—proven to be enough for a valid proof of concept.

## 5. Experimental Antenna Evaluations

The selected devices from Nordic Semiconductor (nRF Series), specifically, the nRF51 dongle and the nRF52840DK board, are a typical representation of a COTS transmitter. For these two devices, we analyzed the performance of their internal built-in antenna (Section 5.1). In our experiments, we measured the wireless communication channel using two methods (Section 5.2) and test external COTS BLE antennas attached to the nRF52840DK board operating as a transmitter.

### 5.1. Antenna Radiation Patterns

The first step is to determine the antenna performance of the internal COTS antennas of the selected nRF51 devices. For the nRF51 dongle antenna, the radiation pattern was determined by measurements in an anechoic chamber. These measurements allow the determination of whether such a COTS on-board antenna is applicable to harsh environment scenarios or whether its performance is simply not acceptable.

For further investigations, we only utilized nRF devices: the nRF52840DK board for connecting and testing the external antennas, and the nRF51 (PCA10031) and nRF52840 (PCA10059) dongles as transmitting nodes. The antenna dimensions of the dongles are depicted in Figure 5.

The antenna radiation pattern of the built-in PCB antenna was measured in an anechoic chamber, for which Figure 6 shows the measurement setup. The results of the measurement are displayed in Figure 7. It is evident that the radiation pattern is nowhere near omnidirectional, which renders it as not recommendable as a receiver for our experimental tests.

In contrast, the monopole antenna of the nRF52840DK board exhibits an omnidirectional pattern. Figure 8 and Figure 9 display the horizontal and the vertical radiation patterns of the nRF52840DK board antenna, which was provided by Nordic Semiconductors [42]. As displayed, the pattern is much more omnidirectional in comparison to that of the nRF51 dongle antenna (see Figure 7), but the maximum gain in any direction is significantly lower.

Taking into consideration the radiation patterns of these two devices, it was decided to perform the experiments using the nRF52840DK board for the receiver device—as its monopole antenna exhibits an omnidirectional radiation pattern.

### 5.2. Measurement of Wireless Communication Channel

For a given wireless setup scenario—using for the transmitter an external COTS BLE antenna in a specific position in the testbed—we evaluate the wireless communication channel by means of two different measurement methods: RSSI and a VNA for S-parameter measurements.

We discuss in Section 5.2.1 whether the RSSI values provide a reliable indicator to derive information about the antenna performance at certain locations of the experimental testbed (for an overview of interactions between antennas and a harsh propagation environment, the reader is directed to Section 3). VNA measurements are required for verification and comparison with the simulation (Section 4).

#### 5.2.1. Received Signal Strength Indicator

RSSI values are just rough estimations and according to the official specifications; the results are merely indications of the received power level by an antenna [43]. There are no means of knowing what happens to the transmitted signal during the entire communication path solely with RSSI.

The RSSI values vary for different devices and are dependent on the utilized communications microchip. For instance, according to the product specifications of the deployed nRF52840 DK boards [44], the resolution of the RSSI is as low as 1 dB, for a range of −20 dBm to −95 dBm @ 1 Mbps (and even −103 dBm @ 125 kbps).

For establishing the RSSI values, the C code of the nRF devices was adapted. The ble_app_uart example from the Nordic SDK (version 17.0.2) [45] served as a starting point. For additional evaluation purposes, the packet error rate (PER) was implemented, which is based on [46], from which the bit error rate (BER) is calculated as well.

#### 5.2.2. Vector Network Analyzer Measurements

In contrast to RSSI values, much more precise measurements can be achieved with a vector network analyzer (VNA). With the VNA, it is possible to analyse incoming and outgoing high-frequency (HF) signals from a device under test (DUT). Those signals are, on the one hand, the ones that are directly transmitted by the DUT, but, on the other, are also reflected signals. For this purpose, the so-called scattering parameters (S parameters) are measured to identify the characteristics of the antennas and the channel.

Recording the scattering parameters with the VNA is the best method to generate data for comparison with simulation. For this purpose, we captured the reflection coefficient $S_{11}$ of the antenna to obtain the information about how much power is reflected from the antenna to see if it is working as expected in the environment. In many cases, there will be frequency shifts or other effects due to the metallic properties of the environment. The reflection coefficient $S_{11}$ characterizes the antenna matching to 50 Ω, while the transmission coefficients $S_{12}$ and $S_{21}$ characterize the channel transfer function. $S_{21}$, known as forward voltage gain, represents the power ratio that is transferred from port 1 (transmitter) to port 2 (receiver). With the measured S parameters, we can gather sufficient information for verification of the simulations.

### 5.3. Additional Requirements for Measurements

The nRF52840DK boards have a very specific SWD/SWF connection, where solely the Murata MXHS83QE3000 measurement probe breaks the circuit of the internal onboard antenna. Only then, the internal antenna will not contribute to the wireless transmission. The connector and the probe MXHS83QE3000 add losses to the RF signal, which, for the frequency of 2.44 GHz, are approximately of 1 dB, according to the user guide [47].

Most of the selected antennas have a matched 50 Ohm cable with a u.FL/I-PEX MHF radio frequency (RF) connector attached to it. Since those connectors are fragile and might break after a few plug-in motions, the sensible solution was to create an adapter (SMA to u.FL) for each antenna, so that this bottleneck is avoided. Such an adapter (Figure 10) was manufactured for each tested antenna and calibrated to achieve the matching at 50 Ohm input impedance and optimum EM radiation power. The losses at the SMA connector for the MXHS83QE3000 are significantly lower than using an SMA plug to MHF jack adapter (e.g., ADP-SMAM-UFLF [48]).

## 6. Experimental Results

The focus of our data collection and its analysis is to test whether the proposed EM simulation setup delivers accurate-enough predictions about the qualities of the wireless communications in our experimental testbed to provide antenna recommendations. Our results and discussion are in the context of our proof-of-concept setup, that is, our experimental testbed (Section 6.1) and its simulation counterpart (as seen in Figure 4 and Section 6.7). Hence, our analysis is representative of real in-vehicle behavior to the degree that our setup reflects electromagnetic characteristics sufficiently comparable to those of a real engine compartment.

### 6.1. Experimental Setup

Due to the nature of the final envisaged implementation, an initial assumption is that the receiver’s position is fixed, where the RX WSN is installed at a fixed position in the dashboard of a vehicle. The transmitter or TX WSN, however, can be positioned anywhere within the experimental testbed—or anywhere in the engine compartment. We always used the nRF52840DK board with its onboard monopole trace antenna as receiver (as seen in Figure 2), which achieves an omnidirectional radiation pattern—meaning that it receives the EM waves comparably well in all incoming ray directions. A second nRF52840DK board—to which an external COTS BLE antenna is connected—was used as transmitter.

For the experiments, we used the following settings in our BLE setup (see Table 3). A transmission power of 0 dBm— which could be as low as −20 dBm with the nRF52840 device for RSSI measurements [44]—was kept consistently the same for all experiments to attain consistent results. Measurements are carried out in the 1 Mbps mode since it satisfies the requirements of most real application scenarios in the automotive context.

An overview of the selected measurement positions for the transmitter is provided in Table 4 and Figure 11. A list of the 21 utilized COTS BLE antennas is provided in Table 5.

We utilized the VNA ZVA 24 from Rhode & Schwarz for data acquisition in our experiments. A ZV-Z32 PC 3.5 Fixed Match Calibration Kit was used to calibrate the device. The entire setup with the test cables is displayed in Figure 12. At the input ports of the ZVA24, 3.5 mm Rohde & Schwarz GS ZV-Z93 test cables were mounted. Additional low-loss RF SMA cables were used as an extension to reach the defined positions in the setup. The bend radius of the cables is within the specification limits to keep losses to a minimum, which were then negligible.

### 6.2. Preliminary Experimental Analysis

In the implemented evaluations, the impact of the various positions was investigated by means of measurements. We present an overview of the experimental results expected and achieved at each of the selected positions in Table 6. Each position in the setup delivers different results, based on distance to the metal, antenna type, position, and orientation of the antenna.

First, information was accumulated based on experimental measurements only. The RSSI was used to identify interesting scenarios for further analysis and the VNA was used to verify the RSSI values. As discussed in Table 6, position 1 and 3 showcase the best conditions to achieve good wireless communication conditions. By working on these two positions, we obtained the optimal dongle-to-antenna power load matching to achieve maximum transmission power. The results repeatedly showed good consistency with reliable communication according to our setup and the information provided on the antenna datasheets.

Second, in order to keep the amount of experimental effort manageable while still extensive enough to deliver a solid research analysis, positions 3 and 4 were selected for further study. For position 4, the direct line of sight between transmitter and receiver is completely blocked, while at position 3, the direct path is not fully blocked. For position 3, the antennas performed somewhat similar to the free space scenario, even in the presence of the object. This circumstance explains why position 4 is the most interesting one for our analysis, and the reason why our presented results focus almost exclusively on it.

Our work on position 4 proceeded as follows. Data acquisition of RSSI values and with the VNA was performed and used to assess the quality of the RSSI values in a harsh wireless communication scenario (Section 6.3). Then, EM simulations were performed and the S parameters derived from the VNA measurements. A comparison between both was carried out to verify the EM simulations by analyzing the results of 14 simulations (Section 6.4). The challenges of EM antenna modeling are discussed in Section 6.6. In Section 6.5, the case for heterogeneous COTS antenna setups is discussed based on VNA measurements and, afterwards, EM ray-tracing simulations were performed for the corresponding scenarios, for which we show six simulation results (Section 6.7).

### 6.3. RSSI versus VNA Measurements

Staying on the same scenario—Antenova SR4W030 antenna at position 4 with the lid of the experimental testbed shut—our last observation can also be investigated with basic RSSI values. For this purpose, Figure 13, Figure 14, Figure 15 and Figure 16 display the RSSI values for the two mentioned scenarios. In Figure 15 and Figure 16, the antenna is rotated 90 degrees to investigate differences and influences on the antenna characteristics. It is noted here that, since the testbed’s lid is shut, the experimental testbed can be considered as a Faraday cage and external influences cannot impact the measurements.

In an unobstructed scenario, where the antenna is placed on the front panel of the physical setup (position 4), we expect a good behavior since, according to the data sheet [52], this specific antenna (SR4W030) should not detune close to metal. This is clearly visible in Figure 13 with reference to the RSSI values, since they are close to an optimum (which lies at approximately −16 dBm for these nRF devices).

When the metallic object is present and obstructing the direct line-of-sight communication path (Figure 14), an impact on the RSSI values is clearly visible. For some channels, the RSSI indicates a strong deterioration in the transmission behavior. This influence can be observed when performing several measurement runs with different antennas, where the offset in the results for the same position is sometimes quite high. In most cases, these strong variations can be traced back to the antenna characteristics that are influenced by metallic surfaces in the vicinity.

When the antenna is rotated 90° and the object is not present (Figure 15), the overall performance of this specific antenna is reduced—compared to the non-rotated case (Figure 13). The antenna structure is now normal to the front plane and the metal clearly impacts the near field of the antenna. Some channels on the BLE frequency spectrum still work rather well, while others were clearly affected. This type of behavior can indicate a shift in the frequency antenna characteristic, but the exact reason cannot be diagnosed using RSSI values only.

When the antenna is rotated 90° and the metallic object is blocking the direct line-of-sight communication path (Figure 16), the overall performance is reduced—compared to the non-rotated case with object. However, when compared to the rotated case (Figure 15), surprisingly, some of the channels actually worked better.

In comparison with another antenna type, e.g., the Taoglas WCM.01.0151W rotated 90° and the metallic object blocking the direct line-of-sight communication path (Figure 17), the wireless link operates almost as if the object was not there (compare RSSI values of Figure 17 against those in Figure 16). This behavior is caused by the favorable WCM.01.0151W antenna’s radiation pattern for this situation.

In Table 7, we show the measured RSSI values and the S-parameter $S_{11}$ values inside the BLE frequency band derived from VNA measurements for each scenario: ‘NO’ no object or ‘O’ object is present inside the toolbox; and the lid is either ‘LO’ open or ‘LS’ shut (closed). The antenna column indicates the antenna that was connected to the transmitter device. The values between columns representing the same scenario are color-coded in {GREEN, YELLOW, ORANGE, RED} according to the assessed quality of the wireless communication—the thresholds for the color codings are shown in the last four rows of the table. Studying how antennas fell into the assessed-quality color-coding groups leads to interesting insights.

In Table 7, all antennas in the GREEN group of the VNA S parameters are in the GREEN group according to the RSSI values. Similarly, 90% and 100% of antennas—for ‘NO-LS’ and ‘O-LS’ respectively—in the GREEN or YELLOW groups of the VNA S parameters are in the GREEN or YELLOW groups according to the RSSI. This is sensible, since the VNA measurements have more predictive power for the quality of the wireless link than the RSSI.

In Table 7, 80% and 86% of antennas—for ‘NO-LS’ and ‘O-LS’ respectively—in the ORANGE and RED groups of the RSSI values are in the ORANGE or RED groups according to the VNA S parameters. This is sensible, since the RSSI values reflect that the quality of the wireless link is, in this case, bad, which is then well-observed with the VNA.

In Table 7, 69% and 64% of antennas—for the ‘NO-LS’ and ‘O-LS’ scenarios respectively—in the GREEN or YELLOW groups of the RSSI values are in the GREEN or YELLOW groups according to the VNA S parameters. Considering that the VNA measurements are more precise, this observation shows that using the RSSI values to select antennas may lead, in 31% and 36% of the experiments, to an antenna that is ill-suited for the corresponding scenario. Therefore, based on these data, using the RSSI values for antenna recommendations may be not recommendable.

As shown in this section, in some cases, metallic objects blocking the direct line-of-sight communication path can even have positive effects on the transmission behavior of antennas. Positive interactions between antennas and neighboring metallic objects happens either if the distance to the ground plane is a multiple of the wavelength or if the field propagation is reinforced by the metal itself. In general, the reason for antenna behavior differences—either deteriorations or improvements in the wireless link—cannot be clearly diagnosed from parameters such as the RSSI values, since they provide no real information about the frequency response of the antenna ($S_{11}$ curve)—each channel occupies a small band inside the BLE frequency band—or any effects at the transmitter side.

In contrast, the EM simulation and the VNA measurements allow the derivation of the $S_{11}$ frequency response curve of the antenna for a given setup, thereby providing much more predictive power. Analyzing the $S_{11}$ curves in Figure 18 provides a diagnosis about the root cause of effects observed in the RSSI values measured for the same situation. To illustrate this, we can analyze the corresponding figures and curves; Figure 18 corresponds to Figure 13 and Figure 18 to Figure 14. Comparing both VNA-based $S_{11}$ curves shows that the presence of the object is both causing a shift in the frequency response of the antenna ($S_{11}$ curves shifted rightwards) and reinforcing the antenna behavior for one of the BLE channels (small dip in $S_{11}$ at the beginning of the BLE frequency band).

As discussed in Section 6.4, due to licensing and computing HW limitations, we cannot—at the current time—obtain the $S_{11}$ curves from EM simulations with a finer frequency sampling. Therefore, small dips in the $S_{11}$ curve are not resolved in our EM simulations. However, we have often been able to identify shifts in the antenna frequency response, which have then always—for all tested cases—been reflected on the VNA-based $S_{11}$ curves.

Ultimately, such direct experimental comparisons—as the ones presented in this section—can indeed be performed using the RSSI values, but they require the following careful experimental procedures. The number of experiments required increases significantly with the number of candidate antennas and positions in the testbed—as well as with the number of orientations of the antenna in each position. In contrast and in our experience, in the vast majority of cases, the RSSI values correlate with the VNA-based $S_{11}$ curve and its EM simulation counterpart in terms of detecting whether an antenna is going to work well for a specific position or not. These facts are the main motivators for our development of a prototype implementation of an automatic antenna-recommender system as part of the presented research work.

### 6.4. S Parameters EM Simulation versus VNA

In this section, we compare the S parameters from the EM simulation results with those derived from the VNA measurements. When considering our wireless communication scenario, what matters is the value of the S parameters inside the BLE frequency band (2.402–2.48 GHz)—see Figure 18 for an example.

First, we checked whether both S parameters curves are reasonably close. Table 8 shows data of the S-parameter $S_{11}$ curves derived from the VNA measurements with those from the EM simulation results, for experiments performed in the same scenario: ‘NO’ no object or ‘O’ object is present inside the toolbox with the ‘LS’ lid shut (closed). The antenna column indicates the antenna that was connected to the transmitter device. The values between columns representing the same scenario are color-coded according to the difference in the absolute value of $S_{11,max}$ using the thresholds {3 dB, 7 dB, 10 dB} (see Table 8’s caption). For the {‘NO-LS’, ‘O-LS’} scenarios there are {5,6,2,1} and {13,1,0,0} values, respectively, in the {GREEN, YELLOW, ORANGE, RED} error ranges. This effectively demonstrates that the error levels are reasonable for the ‘NO-LS’ and negligible for the ‘O-LS’ scenarios. Therefore, we conclude that the EM simulation results are validated by the VNA measurements.

In addition, secondly, we analyzed the interesting case of the Antenova SR4W030 antenna at position 4 (see Figure 11), for which the S parameters are plotted in Figure 18. The lid of the testbed was shut in this experiment—this is the case for every experiment that was compared to EM simulations, as our EM simulation model is designed for the lid-shut case. There was no object within the setup in the first evaluation run, while in the second run, a metallic object was positioned in the direct communication path. In the second scenario, the direct line-of-sight communication path is blocked, and the signal can only reach the receiver by undergoing 1 or more reflections.

An inspection of the curves for the S-parameter $S_{11}$ (Figure 18) shows the sampled frequencies are different and less spaced in the EM simulation. This is because the frequency sampling performed by the EM simulation is intrinsically decided by the solver depending on the intermediate results of the iterative process, while the final total number of frequency samples is limited by our usage of a basic license of CST Studio Suite 2021 and our available computing hardware. In spite of these limitations, the S-parameter $S_{11}$ curves (Figure 18) demonstrate that, in both cases—NO and O—there is a strong correlation between the $S_{11}$ curves—from VNA measurements versus EM simulations.

Furthermore, when there is no object in the transmission path, the results indicate a better transmission scenario than when the object is present, which can be observed as a deeper dip on the $S_{11}$ curve inside the BLE frequency band. An additional observation is that—probably due to the short distance from the metallic object to the transmitting antenna in this specific scenario—it can be observed that the achieved results at a frequency of 2.4 GHz are slightly better with the metallic object in the close vicinity, which is observed as a small dip in the corresponding VNA-based $S_{11}$ curve.

### 6.5. Heterogeneous Antenna Setup

It is sensible to apply the gained knowledge to a practical example, which demonstrates the applicability and usefulness of the approach. In this example, the results of two antennas at two different positions (3 and 4) are compared and discussed. The selected antenna candidates are the Antenova SR4W030 and the Laird 001-0015 FlexNotch antenna. Our analysis will focus on the VNA-based S-parameter $S_{11}$ curves of both antennas at the positions 3 and 4, which are shown together in Figure 19 for easy comparison.

By analyzing the depth of the dips of the $S_{11}$ curves in Figure 19, the following can be predicted. First, that the Laird FlexNotch antenna will perform very well at position 3 and is going to be one of the best candidates for this position. Second, that this antenna will fail to achieve a good-quality wireless link when placed at position 4—an effect most probably caused by the very close proximity of the testbed’s metal sheet. Additionally, third, that while the Antenova SR4W030 will not be the best candidate for the unobstructed scenario—or direct line-of-sight situation—(position 3), it can perform considerably well when placed adjacent to metal surfaces (position 4), especially in relation to the performance of other antennas in such conditions.

These results are both confirmed by: (1) the experimental test of the antennas and the acquisition of the RSSI values, and (2) the $S_{11}$ curves obtained from EM simulations.

The comparison presented in this section shows that our approach to identify the best antennas candidates for specific locations, based on the $S_{11}$ curves—which can be derived from VNA measurements or from EM simulations—can effectively predict the behavior of the antennas in locations presenting very different electromagnetic constraints or boundary conditions on the wireless data transmission.

In addition, this section underscores the fact that it is not possible to find a single antenna that will fit the best for all the positions in the harsh propagation environment. This means that there will be several antennas which are best applicable for various positions while other antennas outperform them at other locations—due to the varying EM constraints and boundary conditions associated to each location. In terms of wireless communication optimization, this means that a small number of antennas can be pre-selected for each of the positions to achieve the maximum data throughput and reliability by means of the predicted S-parameter $S_{11}$ curves from EM simulations. Further experimentation in more complex scenarios, such as in real car engine compartments (Figure 3), is necessary to determine the optimum balance between antenna selection or pre-selection through EM simulations and the required number—if any—of further experiments. These facts underline the motivation for the development of the presented approach.

### 6.6. Electromagnetic Antenna Modeling

In this section, we present issues that we encountered that underscore one of the challenges of achieving an accurate EM simulation, namely, the creation of the EM simulation models of the COTS antennas.

As a first example, the VNA-based versus the EM-simulated S-parameter $S_{11}$ curves for the Laird 001-0015 FlexNotch antenna at position 3 are shown in Figure 20. It is rather straight forward to notice a shift in the frequency response of the antenna between the VNA-based measurement and the simulation result. In order to solve this particular issue, the authors needed to test with different substrates for the board material around the antenna, until the correct substrate was found. The presented challenge is a potential disadvantage of the usage of EM simulations to predict the behavior of COTS BLE antennas.

As a second example, the EM-simulated S-parameter $S_{11}$ is shown in Figure 21 for the antenna Taoglas FXP75070045B modelled according to the specifications of its datasheet. In this case, our investigations into the bad performance predicted by the EM simulation result led to the conclusion that the physical dimensions or materials of the antenna specified in the datasheet are not accurate.

Regarding six simulations performed at position 4 to test for hostile communication conditions, the simulation results provided a basis to predict the adverse nature of this position for the COTS BLE antennas. The four shifts on the frequency response of the antennas and the two very weak transmission setups could be identified from the S-parameters—specifically from $S_{11}$—obtained from the simulations. It is noted here that, given our setup, it was almost always possible for the devices to establish a communication link, sometimes with low device self-reported RSSI values. The hostile nature of the communication conditions was further measurable by high amounts of package losses and a reduced communication bandwidth—i.e., a reduced information transfer rate.

An important remark here is that, for the EM antenna modeling (Figure 20), we utilized a frequency domain solver with the high-frequency conducting wall boundary condition instead of the configuration described in Section 4. This type of EM simulation is fundamentally different to the one used to estimate the EM-simulation S-parameter—for instance, the solver focuses numerically on the volume of the environment immediately around the antenna of the transmitter to simulate the near field of the antenna. That is a box in the order of magnitude of 1–2 times the wavelength of the frequency under investigation—the middle frequency of the BLE band—since the antennas are smaller than one-half its wavelength. In effect, this results in significantly faster EM-simulation execution times, that allow, in a relatively short period of time, the iterative improvement in the EM antenna modeling—particularly when compared to the full EM model simulation. This is the reason why the $S_{11}$ curve in Figure 20 and Figure 21 contains many more frequency samples than those in Figure 18.

### 6.7. Ray-Tracing Simulation Results

The previously presented illustration (Figure 4) shows the ray-tracing simulation results for position 4 (without the object), for which the Antenova SR4W030 antenna was the transmitter while the monopole onboard antenna of the nRF52840DK BLE dongle was the acting receiver. This ray-tracing simulation corresponds to an experiment where the direct line-of-sight path is unobstructed.

When adding an object to the same experiment—and the EM simulation setup—the immediate effects become visible in the ray-tracing results—see Figure 22. The plotted rays are the only ones that hit the receiver before becoming invalid—that is, before becoming too weak due to ray length or undergoing too many reflections. The figure clearly shows that the direct line-of-sight path is blocked by the object, and that only reflected rays can arrive at the receiver.

For the two ray-tracing simulations shown in Figure 4 and Figure 22, a maximum of four reflections—as the threshold to define valid rays hitting the receiver—and a minimum arrival power of 40 dBV/m were used. The characteristics of the valid rays are shown in Figure 23 and Figure 24, that is: ray-length, ray power at the point of arrival, and emission angles of the rays with respect to the transmitter.

Figure 23 shows the electric field strength at the receiving device and the percentage of the hitting rays (with a given length) versus the ray length. This percentage is mainly an estimate of the number of valid rays; the higher the percentage, the fewer rays were successful.

As expected, many more rays arrive at the receiving device in the line-of-sight (LOS) scenario and the number of rays as well as the electric field strength is considerably higher than in the scenario with the obstruction—or non-line-of-sight (NLOS)—scenario. One of the main facts that is reflected by the ray-tracing simulations is that the rays lose energy with each reflection—and the characteristics of the reflection—and with increasing ray-lengths. This fact is well-reflected in our ray-tracing simulation results, and it is observable in all four figures, as the rays in the NLOS scenario reach the receiver after traveling longer distances and undergoing more reflections.

Figure 24 shows the emission angles (polar—or inclination—angle, θ, and the azimuthal angle, ϕ) of the rays launched by the transmitter for the LOS (Figure 4) and NLOS (Figure 22) scenarios. An interesting information shown for the LOS scenario is that there are five different regions – or solid angles – through which rays launched by the transmitter are able to hit the receiver; whereas in the NLOS case all but one of these regions are blocked or almost blocked. These facts are also visible in both 3D visualizations (Figure 4 and Figure 22) but more difficult to observe than in Figure 24.

The ray-tracing simulation has the main advantage of providing a second way for specialists to interpret the EM simulation results, aside from the $S_{11}$ parameter curve, and to associate the results more clearly to certain characteristics of the utilized antennas, such as their radiation pattern (Figure 7), which characterizes both their directional EM wave emission and absorption capacity.

For non-specialists, the ray-tracing visualization is understandable, and it transmits them a certain level of expert knowledge. For instance, it displays graphically a part of the reasons that make obstacles reduce the quality of wireless links, the number of displayed rays represent the robustness of the wireless link to certain types of interference—e.g., the presence of metallic obstacles—as well as some basic facts related to EM wave propagation.

In addition, the information obtained from the ray-tracing simulation results can be utilized for further optimization in terms of reliability and throughput in the wireless network. First, further analysis attempted to maximize the electric field strength at the receiving device when more directed antennas are used at the identified transmitter emission angles that achieve successful hits on the receiver and where, simultaneously, the most energy is effectively transmitted to the receiver. Second, the power prediction of the hitting rays provides very useful information when considered together with the signal-to-noise ratio requirements of the setup—which is affected by the utilized communication devices, the communication protocol (e.g., Bluetooth), etc.—in terms of the robustness of the particular communication setup to EM noise present inside the engine compartment. As our research work proves, ray tracing seems to be exceedingly convenient for our engine compartment application, as its simulation results match the physical measurements with remarkable accuracy.

In Table 9, the ray-tracing results for six simulations in three scenarios are displayed that correspond to the antennas used to show the case for heterogeneous COTS antenna setups (Section 6.5). Again, we compared the results at position 4, when there was an object in the communication path and when there was none. Additionally, position 3 was used as further comparison to demonstrate the effect of the object. For both, ray length and ray power, the minimum/maximum length and minimum/maximum power are shown. The power indicates how much electric field energy is left when the ray arrives at the receiver. Q1, Q2, Q3 represent the 25%, 50% and 75% quartile of the acquired values, respectively. The last column of the table shows the ratio of hits to launched rays in percentage—i.e., a percentage of the 1000 originally launched rays.

The facts that, in the unobstructed case, position 4 is harsher for the antennas than position 3, and that the obstacle makes the scenario in position 4 the most challenging, are well reflected by the percentage of rays hitting the receiver in the EM ray-tracing simulations (see Table 9). Regarding the ray length, the shortest paths—as specified by ‘min’ and Q1—are similar for unobstructed scenarios; however, these paths are blocked by the obstacle in position 4. Regarding power, in the unobstructed scenarios the maximum energy is much higher than in the scenario with the object. The obstacle even leads to the circumstance that no launched ray will hit the receiver when the Laird 001-0015 antenna is used for position 4. In position 3, however, the Laird 001-0015 achieves the best results—in terms of power and percentage of launched rays hitting the receiver—which is confirmed by the VNA measurements (see Figure 19 and Section 6.5).

### 6.8. Prototype Antenna-Recommender System

A tool that allows test engineers to try out the developed approach was found to be an effective way to gain insights into the technical problem under investigation. Initially, this tool was a simple web-based tool that allowed analysis of specific pre-tested scenarios and result summaries. Our tool evolved into a prototype antenna-recommender system (Figure 25) which can be used as a demonstrator of our developed approach, which consists of the following components.

A back end—where the calculations (e.g., BER estimation and the antenna recommender algorithm) are performed using Python—contains a database with experimental and simulation results. The database contains pictures of the EM antenna models and the following data per testbed position: datalogs of RSSI, PSR and battery values, and the VNA measurements; the VNA-based and EM-simulation S-parameters; and pictures and statistics of the ray-tracing simulation results.

For a selected testbed position, the antenna-recommender algorithm: (1) takes either the VNA-based or the EM-simulation $S_{11}$-parameter curves for each antenna, (2) extracts the $S_{11}$ data for the BLE frequency band—using linear interpolation when necessary, and (3) achieves the antenna ranking by sorting the antennas according to the strongest dip in their $S_{11}$ data—i.e., the minimum value. This simple antenna-recommendation algorithm allows us to utilize both sources for the $S_{11}$ curves—VNA- or EM-simulation based—in spite of the relatively sparse frequency sampling achieved through the EM simulations.

A web-based front end (Figure 25) allows interfacing and displaying data and results. The tool allows—while connected to the BLE dongle—the observation in real-time of RSSI, PSR, and battery-lifetime estimation values (Figure 26). Alternatively, the user can access the antenna recommendation for a selected position as well as visualize and display data related to this position, such as $S_{11}$-parameter curves and the ray-tracing simulation pictures and statistics (Section 6.7).

The ranking orders for two scenarios in position 4—either using VNA- or EM-simulation-based S-parameter curves—are shown in Table 10. As shown in the table, the shifts in the measurement- versus simulation-based rankings are for 86% and 93% of the antennas below four positions. For the ‘NO-LS’ (alternatively ‘O-LS’) scenario, 4 (6) of the top 5 (6) antennas are in both rankings, with maximum differences in the $S_{11,max}$ values of 4.3 dB and 10.1 dB (6.1 dB and 3.7 dB) in the VNA- and EM-simulation-based values, respectively. It is remarked here that the sorting order can be affected by small discrepancies between simulation and measurement values—thereby causing big ranking shifts—while—at the same time—the values themselves can be relatively close to each other—see Table 8 and its related discussion on the benchmarking of the EM simulation results by means of the VNA measurements.

During the prototype demonstration, where all data—from experiments and simulations—was pre-loaded, test engineers stated that the antenna-recommender tool is very promising.

## 7. Lessons Learned

During the course of this work, certain decisions and insights have supported a faster and more focused research and development.

Starting with a proof-of-concept kept the amount of work manageable and allowed to follow a more gradual analysis process. This was, for instance, the case for the gradual identification of the most promising experiments, which we described in the preliminary experimental analysis section (Section 6.2) and led to an incremental understanding and solving of the more complex scenarios.

Repeatability tests—this is repeating the preparation of the wiring and connections before an experiment and the repetition of the data acquisition—are necessary for this type of work in order to obtain consistent data, because small errors in the acquisition setup—e.g., in the cabling and setup of the VNA endpoints—render an acquisition session useless.

Discussions with test engineers were very helpful in forcing us to revisit the theoretical and practical connections between the utilized methods—e.g., the assumptions made when designing the antenna-recommender algorithm purely based on the $S_{11}$ curves—and have influenced the design of the prototype antenna-recommender system.

## 8. Future Work

Given our accumulated experience through this presented research, we have identified the following potential venues for future work.

As we could show with our proof-of-concept approach, antenna recommendations for various positions can be achieved inside our experimental testbed—a metallic toolbox with a simple geometrical shape. Materials and dimensions are very close to those of a real engine compartment and so are the measurement results. The next steps are to verify the validity of the approach in more complex scenarios by adding real engine and vehicle components to the experimental testbed. The influence of every single component could then be identified and verified, which would not be possible in a setup, where all components are already installed. This would allow us to create an adaptive recommendation system in which the impact of each single component is modeled.

For the continuation of this work utilizing a real engine compartment as an experimental testbed, we would need to use BLE communication devices that are relatively rugged. Building on our accumulated experience on the nRF52840 device family, we decided to ruggedize them. In the engine compartment, using the nRF52840DK boards is impractical, particularly in confined spaces and narrow gaps. For this reason, we are using nRF52840DK dongles, to obtain access to many more positions in the vehicle. When conducting the measurements and during real working mode at a later stage, the devices need to be protected against external influences and it is possible to attach external antennas to the BLE dongles. With some custom modifications and the selection of a fitting encasing, we have already fulfilled these requirements. The finished structure with a BLE dongle can be seen in Figure 27.

The adapted dongles are ready for deployment in an engine compartment of a vehicle. Additional alterations can be made regarding the power supply. There is additional space on the other side of the nRF board to connect a round cell battery (e.g., CR2032) to the USB pins to power the dongle. The introduced attenuation with the SMA connector (Telegärtner J01151A0021) [72] and the SMA to u.FL (UMCC) connector (Molex 073386-0850) [73] is similar to the setup with the nRF52840DK boards and amounts to approximately 1 dB.

The experimentation using a real engine compartment would be best realized by accompanying it with EM and ray-tracing simulations in faster computing hardware and a more advanced software license of the CST Studio Suite. This would allow a continuation of the presented work, for instance, with further studies on the repeatability and reliability of utilizing the EM-simulation S-parameters and ray tracing as partial or complete substitutes of experimental measurements—VNA-based S-parameters and RSSI values—in the real engine compartment scenario.

In a complex wireless multi-sensor—or mesh—network, a vast number of unique positions—potentially including various antenna orientations—may need to be analyzed. Many positions, however, will be relatively similar and results at those positions might correspond with each other. In those cases, it seems plausible to think about an automatic classification of similar positions, to reduce the overall number of simulations—and/or measurements—required in the development and testing of the wireless network.

Beamforming with smart antenna arrays has not yet been considered in this work. However, small array structures could also be used in the context of wireless networks to improve overall reliability and data throughput. Since we have a well-defined setup with fixed sensor positions, this could also be implemented, if the size of antenna arrays is small enough in size [74]. Then, algorithms such as the LLMS could be applied to mitigate the intersymbol interference and enhance reliability [75].

When applying beamforming, the SideLobe Level (SLL) reduction plays an important part and approaches such as Sidelobe Sequential Damping (SSD) are vital to achieve a higher signal-to-interference ratio as well as higher spectral efficiency [76]. In future investigations with antenna arrays, such approaches and methods will be compared with the results in our system model.

## 9. Conclusions

Our conducted investigations were aimed towards the temporary instrumentation of vehicles with additional wireless sensor nodes—for instance, utilized in vehicle testbeds to streamline product design and development or to control the vehicle condition in a routine driving test. In such a scenario, a wireless sensor node system, which is pre-configured and optimized for deployment in a certain vehicle, is probably the cheapest and fastest solution.

These facts motivated our prototype implementation of an antenna-recommender system, which we utilized in demonstrations with project partners. Our antenna-recommender algorithm provided almost the same antenna ranking orders, with usually only a few shifts between neighboring positions in the ranking, when using either VNA-measurements-based or EM-simulations-based S-parameter curves. During a prototype demonstration, test engineers stated that the potential benefits of using such an antenna-recommender tool are very clear.

Our experiments were performed with a focus on COTS BLE dongles and antennas. We showcased that some COTS BLE antennas are unsuitable for placement near metallic surfaces. In addition, only some of the antennas fit the requirements of certain—not far-fetched—positions in the experimental testbed. That is, there is no universal antenna design that will deliver the best result for each position in harsh propagation environments. For multi-sensor setups, using different COTS antennas can be required in order to optimize the data throughput and the overall availability and reliability of the wireless networks, as well as to increase their battery life.

During the investigations, a variety of methods were utilized for the evaluation and verification of the correctness of our execution of experiments and simulations. The designed experimental testbed displays electromagnetic characteristics similar enough to those of a real engine compartment to perform our research on EM and ray-tracing simulations and achieve the prototype antenna-recommender system for specified positions. The conducted measurements deliver clear evidence that our followed approach led to the design of accurate EM models for the antennas and the metallic toolbox—with and without obstacle blocking direct line-of-sight communications—and an accurate full EM and ray-tracing model for the experimental testbed. Furthermore, our presented results indicate that cost-effective EM and ray-tracing simulations can substitute the execution of countless arduous antenna measurements and wireless communication experiments on the experimental testbed. The usage of two types of simulations, one for the EM antenna modeling and another for the full EM modeling, allowed to reduce development time—since the full EM simulation requires much longer computation times per execution run.

Our utilized approach is presumably a feasible way to design an accurate EM and ray-tracing model for a real engine compartment.

## Figures and Tables

**Figure 1 sensors-22-06339-f001:**
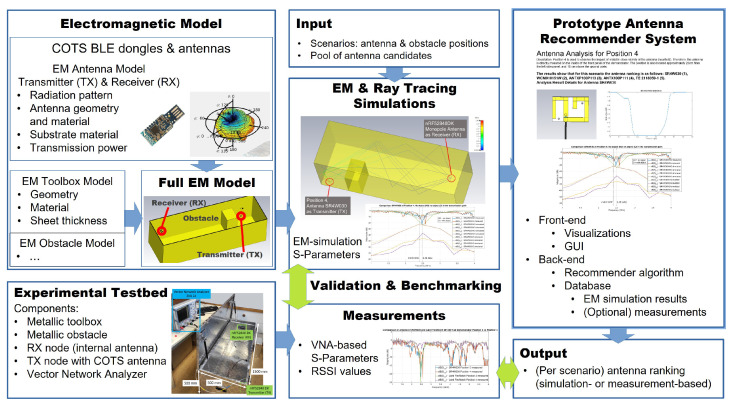
Concept overview of this paper showing the relationship between the prototype antenna recommender system and our research on EM simulations to substitute measurements and the development of an accurate EM and ray-tracing model—utilized for simulating a pair of WSNs on our experimental testbed.

**Figure 2 sensors-22-06339-f002:**
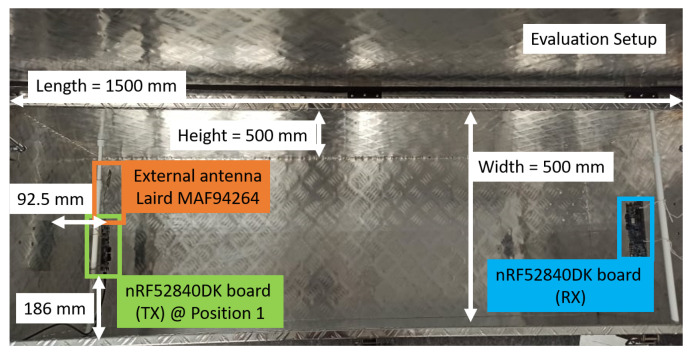
Dimensions of the experimental testbed. In this setup scenario, two nRF52840DK boards are used as WSNs, where the left WSN is acting as the transmitter (TX) at measurement position 1 (see Section 6.1 for details on positions) and the right WSN as the receiver (RX). The distance between the TX and the RX WSNs is approximately 1.25 m.

**Figure 3 sensors-22-06339-f003:**
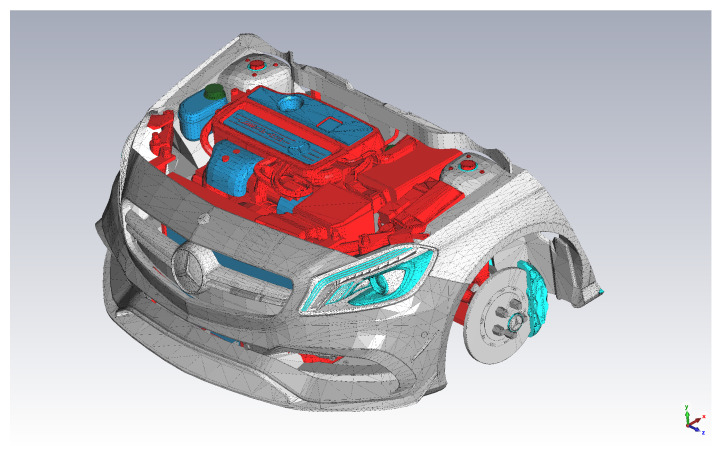
Detailed model of an engine compartment, where material properties are assigned to the various components of the vehicle [41]. The colors are used to highlight different material properties, e.g., red indicates the metallic (iron) components and dark blue indicates plastic covers.

**Figure 4 sensors-22-06339-f004:**
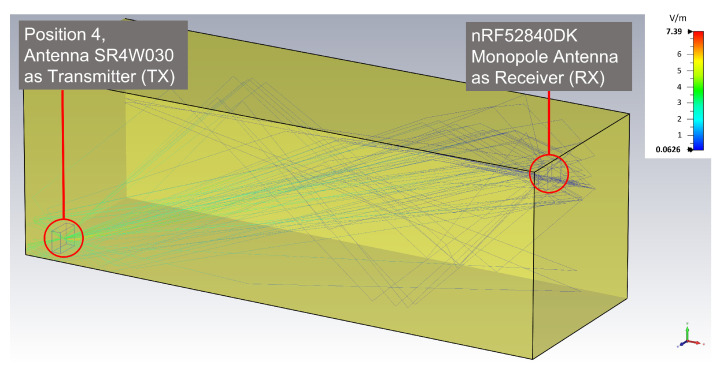
Simulation model of the test environment with material properties and ray tracing between a SR4W030 antenna (TX) and a nRF52840DK monopole antenna (RX).

**Figure 5 sensors-22-06339-f005:**
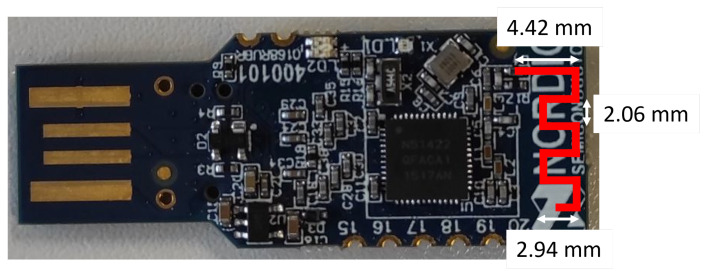
Antenna dimensions of the nRF51 dongle. Such a meandered monopole antenna design is commonly used in wireless communication dongles in the 2.4 GHz frequency band with an input impedance of 50 Ohm.

**Figure 6 sensors-22-06339-f006:**
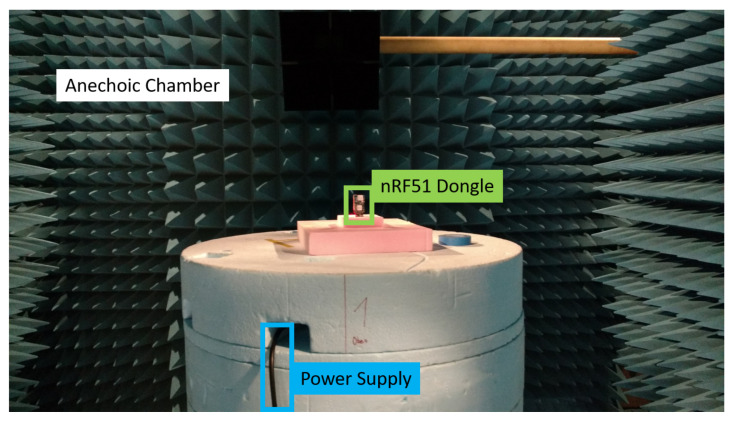
Setup for the measurement of the antenna radiation pattern of the nRF51 dongle in an anechoic chamber. The dongle is positioned on a rotating platform while the power supply of the dongle is managed with a USB cable.

**Figure 7 sensors-22-06339-f007:**
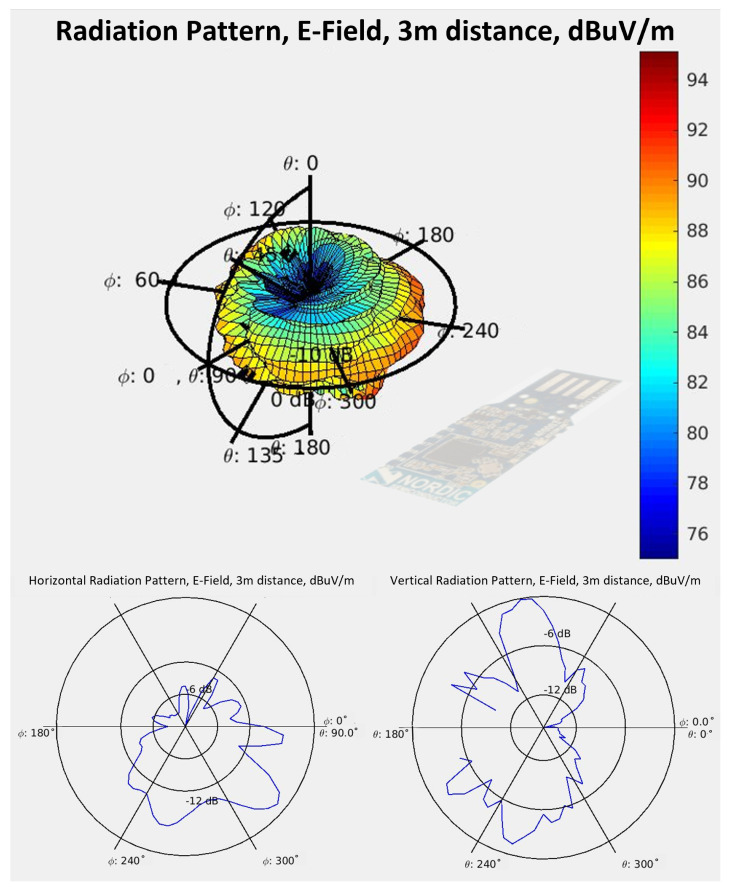
Measured radiation pattern of a nRF51 dongle antenna. (**Top**) 3D illustration of the radiation pattern. (**Bottom left**) horizontal (azimuth) radiation pattern. (**Bottom right**) vertical (elevation) radiation pattern.

**Figure 8 sensors-22-06339-f008:**
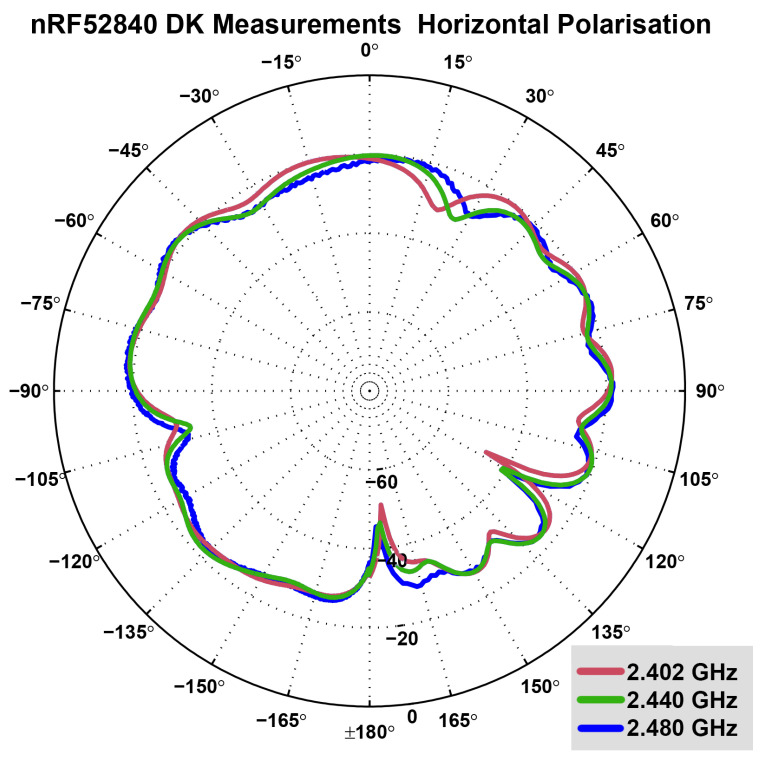
Measured radiation pattern of nRF52840DK antenna: Horizontal (azimuth) polarisation plane with measurements at 2.402 GHz (red), 2.440 GHz (green) and 2.480 GHz (blue). An omnidirectional radiation pattern can be observed, with less radiation at ±180 degrees as well as 120 degrees and −105 degrees [42].

**Figure 9 sensors-22-06339-f009:**
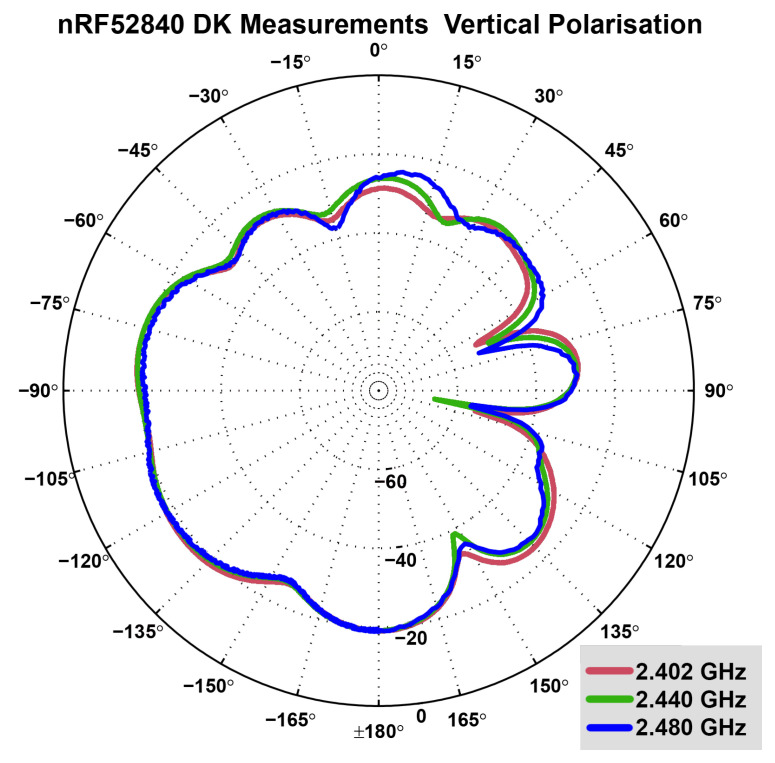
Measured radiation pattern of nRF52840DK antenna: Vertical (elevation) polarisation plane with measurements at 2.402 GHz (red), 2.440 GHz (green) and 2.480 GHz (blue). In this plane, the radiation pattern is distinctly worse in the positive angle area. Radiation at 105 degrees, as well as 70 degrees and 150 degrees is very low [42].

**Figure 10 sensors-22-06339-f010:**
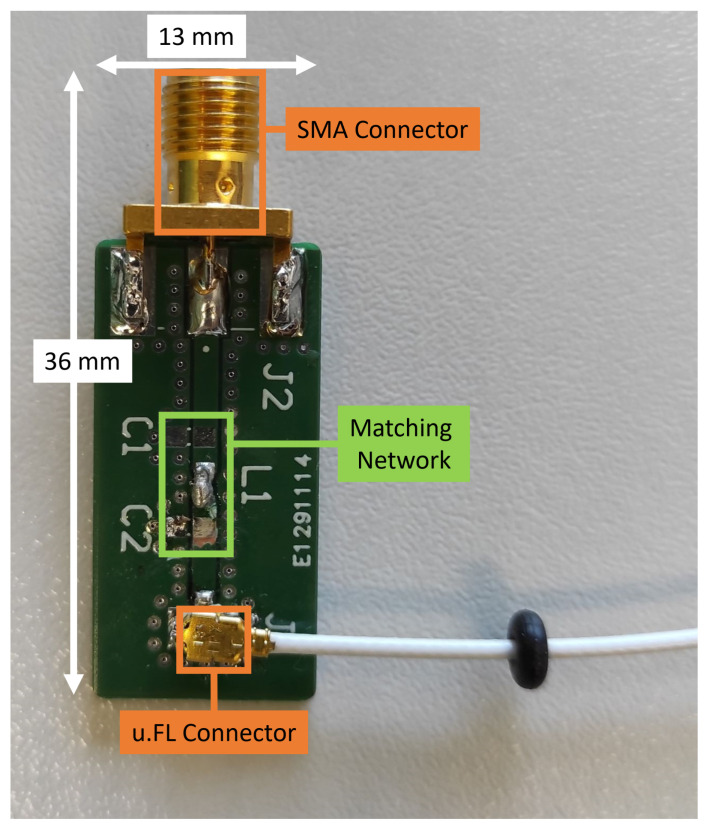
Adapter that was used for each individual u.FL antenna connection, where the Hirose connector U.FL-R-SMT-1(10) [49] is utilized. The SMA connector (Cinch Connectivity Solutions 142-0701-801 [50]) is much more sturdy when connecting/disconnecting. The C1, L1, C2 matching network allows matching the antenna to 50 Ohm.

**Figure 11 sensors-22-06339-f011:**
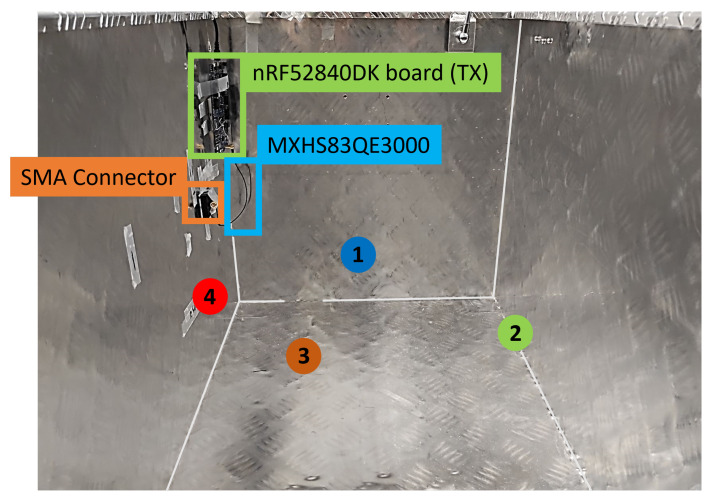
Overview of the selected evaluation positions. Positions {1 (blue), 2 (green), 3 (brown), 4 (red)} show where the TX antenna was positioned in the various measurement scenarios.

**Figure 12 sensors-22-06339-f012:**
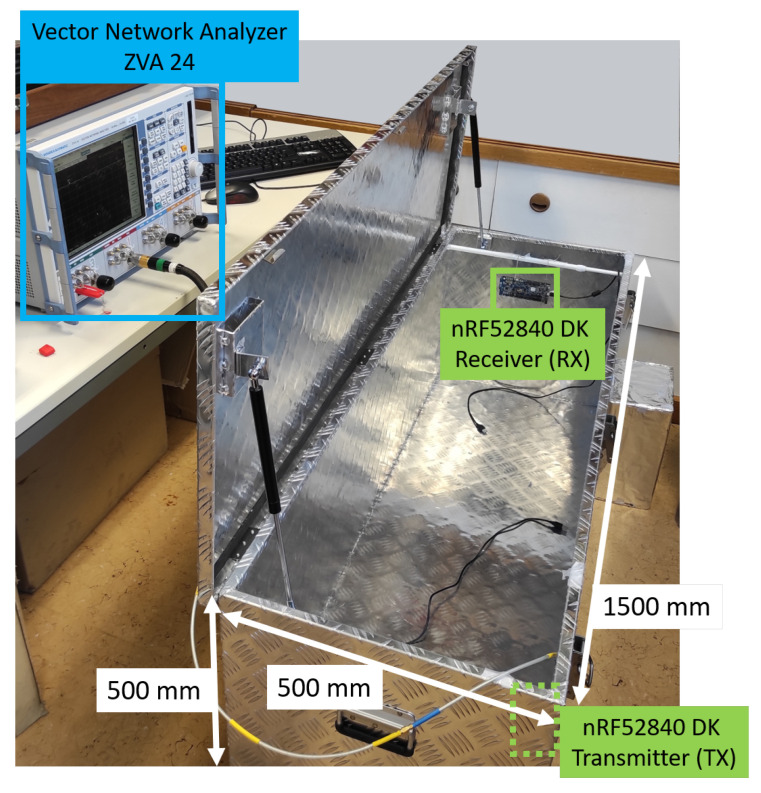
Experimental testbed, physical test setup for measurements with Rhode & Schwarz Vector Analyzer ZVA24.

**Figure 13 sensors-22-06339-f013:**
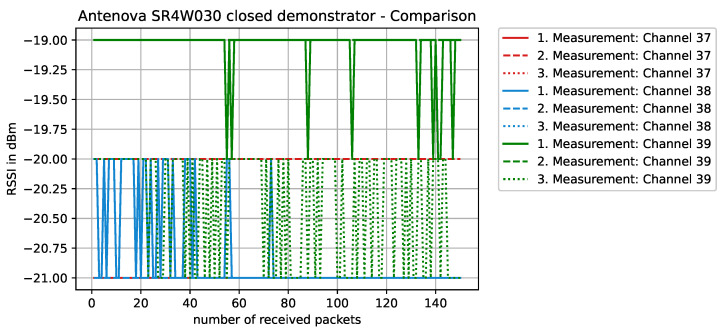
RSSI values with antenna SR4W030 at position 4 with the experimental testbed’s lid shut.

**Figure 14 sensors-22-06339-f014:**
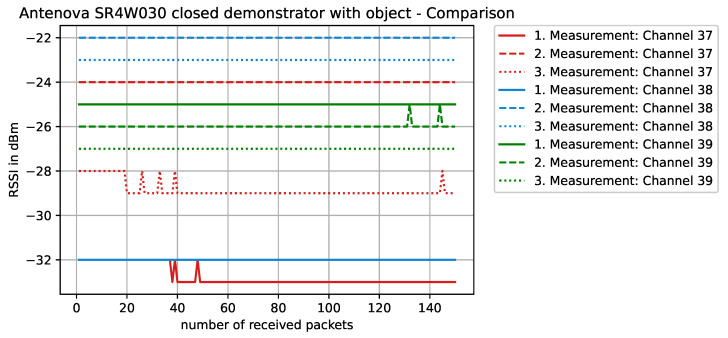
RSSI values with antenna SR4W030 at position 4 with the experimental testbed’s lid shut and the object blocking the direct line-of-sight path of communication.

**Figure 15 sensors-22-06339-f015:**
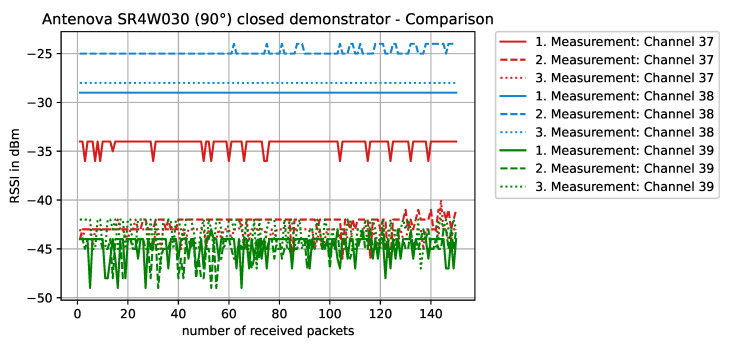
RSSI values with antenna SR4W030 at position 4 with the experimental testbed’s lid shut. The antenna is rotated 90° and the antenna structure is normal to the front panel.

**Figure 16 sensors-22-06339-f016:**
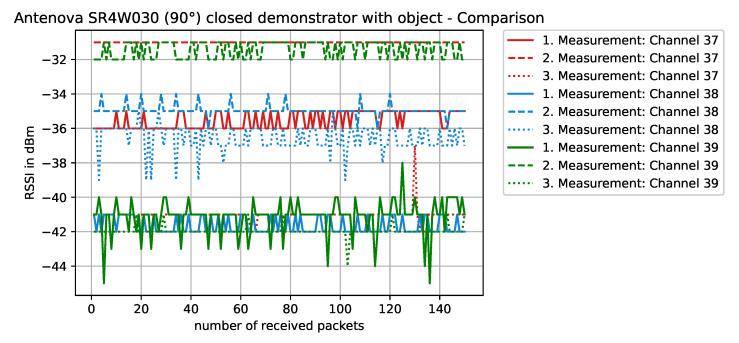
RSSI values with antenna SR4W030 at position 4 with the testbed’s lid shut and the object blocking the direct line-of-sight path of communication. The antenna is rotated 90° and its structure is normal to the front panel.

**Figure 17 sensors-22-06339-f017:**
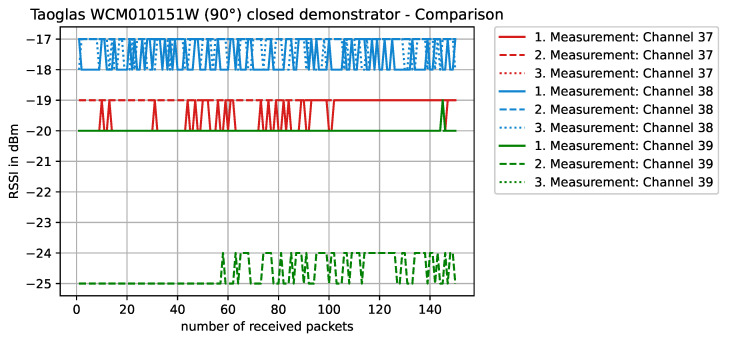
RSSI values with antenna WCM.01.0151W at position 4, rotated 90°, with the testbed’s lid shut and the object blocking the direct line-of-sight path of communication. In comparison with antenna SR4W030 (Figure 16 and according to the RSSI values) it is visible that, in this case, the WCM.01.0151W antenna performs significantly better.

**Figure 18 sensors-22-06339-f018:**
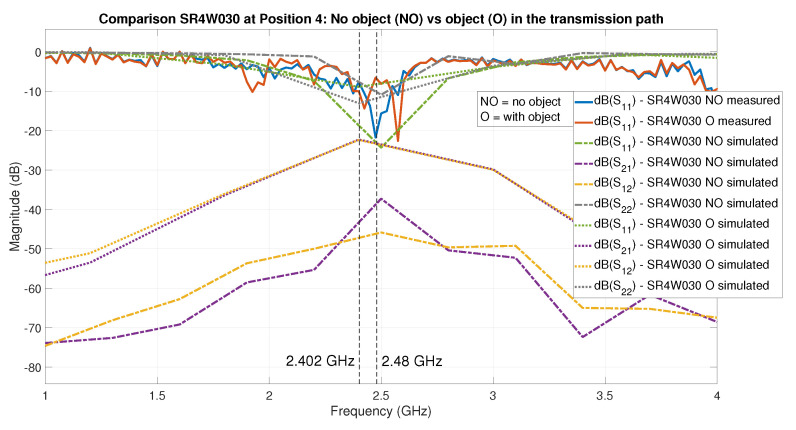
Antenova SR4W030 antenna at position 4, comparison of S parameters from the simulation results with those derived from the VNA measurements. The measurement for $S_{11}$ was recorded with the VNA and shows the results with object (O) and without object (NO) in the transmission path. The measurements are put in relation to the S-parameter results of the EM simulations with object (O) and without object (NO). The area of interest is the frequency band of the operation of BLE (2.402–2.48 GHz), which is marked with dashed vertical lines.

**Figure 19 sensors-22-06339-f019:**
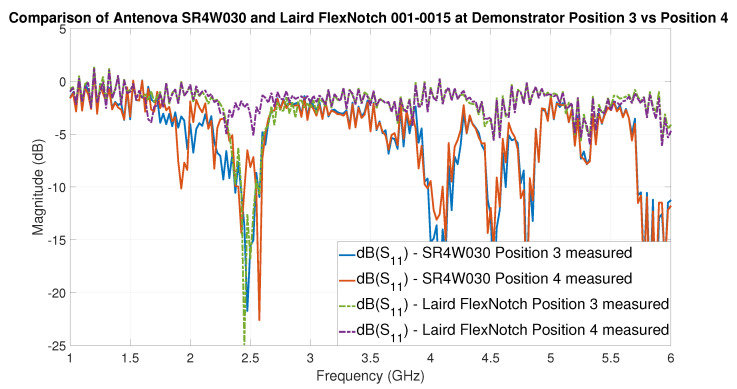
VNA-based S-parameter $S_{11}$ of the two antennas Antenova SR4W030 and Laird 001-0015 FlexNotch at positions 3 and 4 with the testbed’s lid shut. Remark: the frequency band of operation of BLE is (2.402–2.48 GHz).

**Figure 20 sensors-22-06339-f020:**
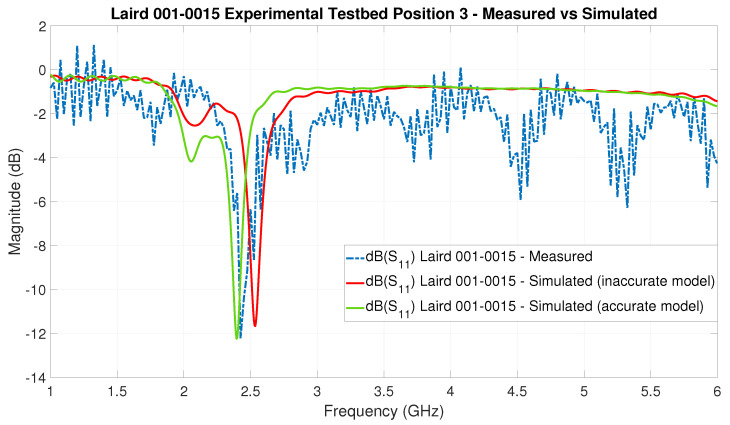
Comparison of VNA-based versus EM-simulated S-parameter $S_{11}$ curves for the antenna Laird 001-0015 FlexNotch at position 3 with the testbed’s lid shut. The two simulation curves showcase the impact, on the EM simulations, of the EM antenna model when not accurately modelled. The red curve shows a frequency shift in the S-parameter $S_{11}$ caused by an incorrect antenna substrate in the model.

**Figure 21 sensors-22-06339-f021:**
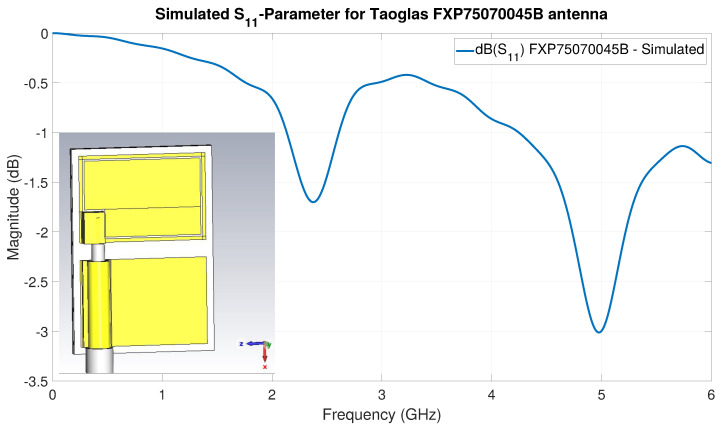
S-parameter $S_{11}$ curve for the antenna Taoglas FXP75070045B—a dual frequency antenna—obtained from an EM simulation. The dips of the curve inside the BLE frequency band and around the 5GHz band are so small that they are an indication of incorrect or inaccurate specifications in the antenna datasheet.

**Figure 22 sensors-22-06339-f022:**
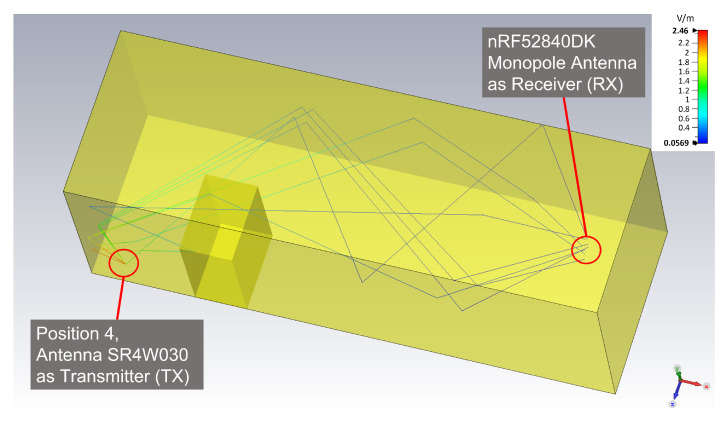
Ray-tracing simulation with metallic object obstructing in direct line-of-sight communication path.

**Figure 23 sensors-22-06339-f023:**
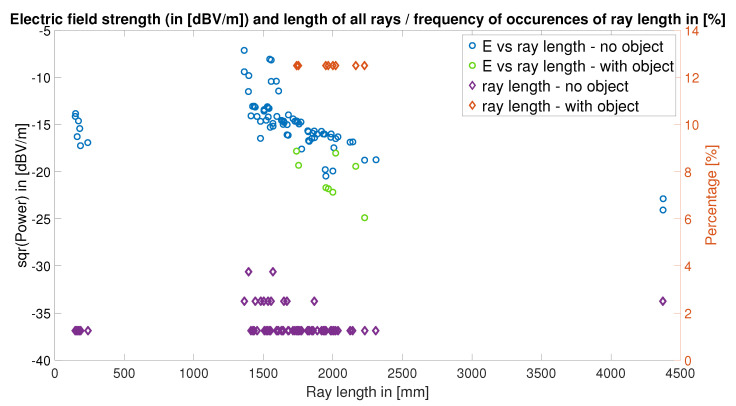
Comparison of the two ray-tracing scenarios from Figure 4 (LOS) and Figure 22 (NLOS). The Antenova SR4W030 antenna is used as TX, the nRF52840DK with the monopole antenna as RX. This scatter plot depicts the length and the electric field strength (in dBV/m) for each of the individual rays. Additionally, the frequency of occurrences (in %) of ray length is visualized.

**Figure 24 sensors-22-06339-f024:**
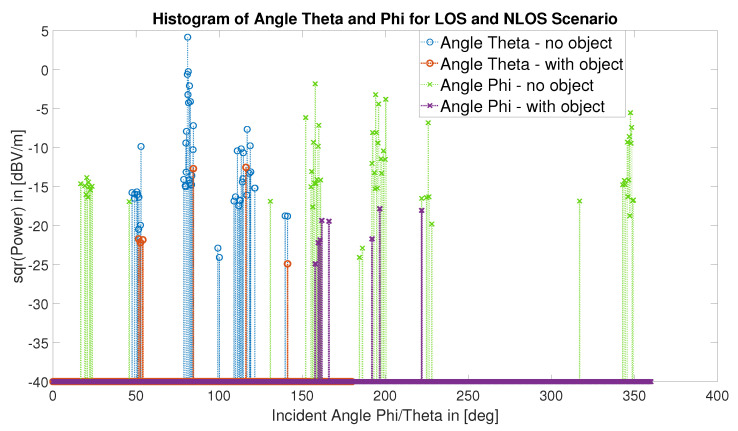
The emission angles (polar—or inclination—angle, θ, and the azimuthal angle, ϕ) of the rays launched by the transmitter are shown for the experiments shown in Figure 4 and Figure 22. The *y*-axis indicates the electric field strength (in dBV/m) which is calculated at the receiver. When considering the angle, from which the rays arrive at the receiver, the angles with the highest transmitted energy can be identified. The higher the field strength, the better.

**Figure 25 sensors-22-06339-f025:**
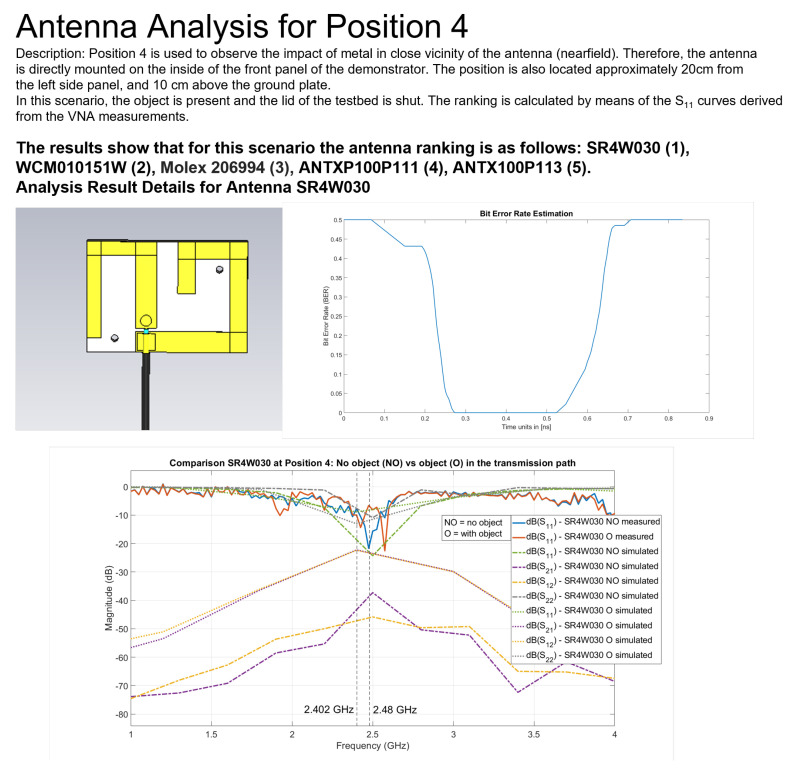
Web-based front-end of the prototype antenna-recommender system, which provides test engineers, for a selected scenario—or position in the experimental testbed—with a visual output of key results and parameters (see Figure 18 for a detailed description) as well as the current antenna ranking based on the data available on the back-end’s database.

**Figure 26 sensors-22-06339-f026:**
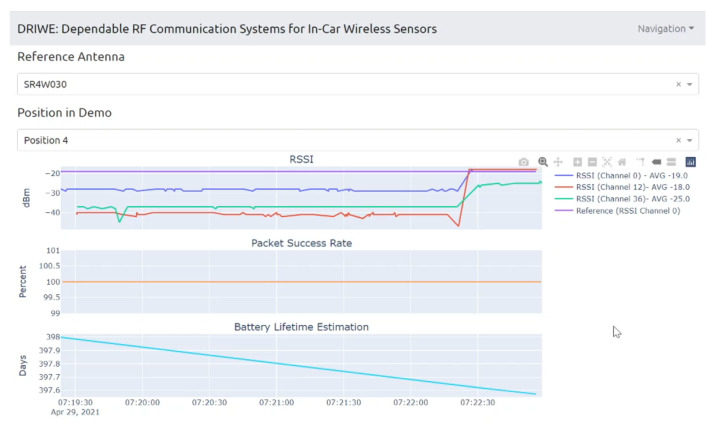
Web-based front-end of the prototype antenna-recommender system, visualization of experimental values for quick evaluation by test engineers. The scenario, antenna, and position are selectable through drop-down menus. The top graph displays the measured RSSI values in real-time in dBm. The middle graph shows the packet success rate and indicates whether a retransmission was necessary. The bottom graph shows an estimation of the battery lifetime.

**Figure 27 sensors-22-06339-f027:**
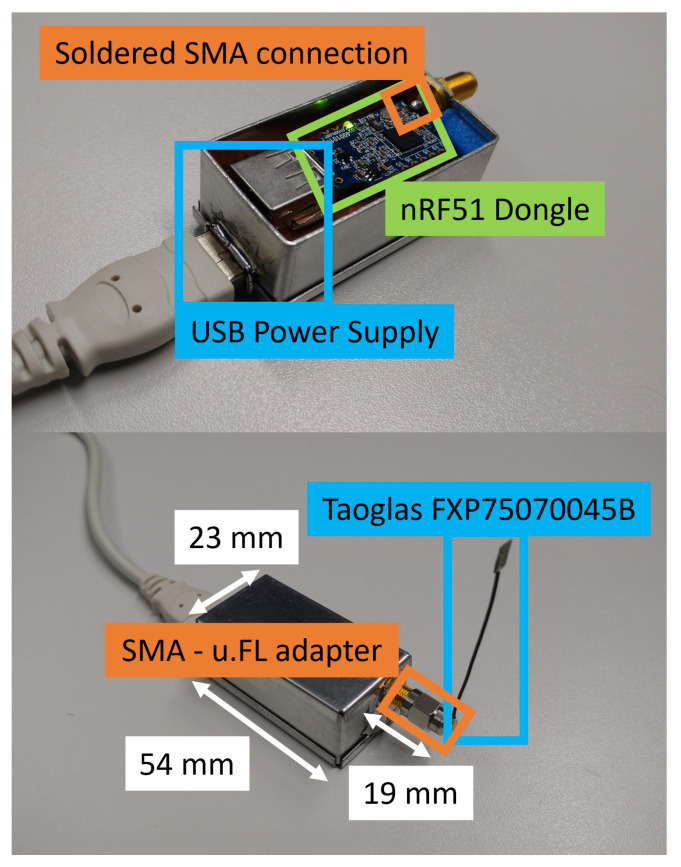
nRF52840 Dongle with open EMC protection case. In the top picture, the dongle is seen with an active USB power connection. The SMA connector is soldered to the matched circuit (50 Ohm) of the cut antenna stub. In the bottom half, a PCB antenna (Taoglas FPX75070045B) is connected to the SMA connector. The encasing can be closed to protect the device from external influences.

**Table 1 sensors-22-06339-t001:** Comparison of previous investigations of in-vehicle RF scenarios. “n/s” is used when no data or sample rates are stated in the referenced work.

References	Design Strategy	Devices	Protocol/ Frequency	Data/Sample Rate	Investigated Range	Metal Nearby
[20]	Tire-pressure monitoring system (TPMS)	Valve stem (TX) Printed monopole (RX)	315 MHz	n/s	Up to 1.6 m	Yes
[21]	Vehicle body effect on TPMS	TPMS with whip, Loop antennas	310/433 MHz	n/s	Up to 3 m	Yes
[22]	Tire-pressure monitoring RFID tags	Dipole/monopole antennas RFID reader and tags VNA	UHF 866 MHz ISM 2.45 GHz	n/s	Up to 2 m	Yes
[28]	Coexistence Zigbee and BLE	CC2520 Zigbee CN512 Bluetooth	Zigbee, BT 2.4 GHz	250 kbps 1, 2, 3 Mbps	Up to 3 m	Yes
[29]	Wireless base station for in-vehicle network	CC2520 Zigbee CC2540 Bluetooth	Zigbee, BT 4 2.4 GHz	250 kbps 1 Mbps	Up to 3 m	Yes
[30]	RSSI-based analysis for intra-car sensor monitoring	HC-05 BT module Smartphone	Bluetooth (BT) 2.4 GHz	38,400 bps	Up to 4 m	Yes
[13,14]	WSNs in heavy vehicles	JN5168 USB dongle	IETF 6TISCH, 2.4 GHz	250 kbps	Up to 6 m	Yes
[16]	RF propagation measurements	Agilent N5182A Agilent N9010A	433/868/915/ 2400 MHz	1 Msps	Up to 2.3 m	No
[15]	Channel loss measurements	USRP B210	Zigbee, BT 5, Wi-Fi 2.4 GHz	20 Msps	Up to 3 m	No
[31]	Characterization of the in-vehicle wireless channel	Agilent E4440A, Schwarzbeck 9113 antenna	2.45 GHz	n/s	Less than 2 m	No
[32]	UWB-IR measurements in in-vehicle scenario	Agilent 8722ET	UWB 3-10 GHz	n/s	Up to 3 m	No
This work	Antenna recommendation of COTS devices in harsh environments	nRF51/52 dongle nRF52840 DK R&S ZVA24	Bluetooth Low Energy (BLE 5.0) 2.4 GHz	1 Mbps	Up to 2 m	Yes

**Table 2 sensors-22-06339-t002:** Settings for asymptotic solver simulation.

Parameter	Setting	Value
Mode	Field Sources	fs2_(RX)1 *
Frequency sweep	Frequency range	2 to 3 GHz
Custom Accuracy	Max. intersections	4
Ray Storage	Visualization options	Show Rays trapped in structure

* Calculation of the near field sources required for the transmitter–receiver communication line and its rays, where fs2 is the transmitting source and (RX)1 the receiving one.

**Table 3 sensors-22-06339-t003:** Overview of settings and key characteristics of Bluetooth Low Energy (BLE) [51] in the experiments.

Parameter	Symbol	Value
Modulation Type	GFSK	data
Bit Rate	$R_{b}$	1 Mbit/s
Bandwidth	B	2 MHz
Transmission Power	$P_{TX}$	0 dBm
Frequency Range		2.402 GHz to 2.48 GHz
Frequency (Advertising Mode)		CH 37: 2.402 GHz, CH 38: 2.426 GHz
		CH 39: 2.480 GHz
Frequency (Connection Mode)		CH 0–39: 2.402 GHz–2.480 GHz
Antenna Gain	*G*	antenna specific (in dBm) *

* the gain of each antenna is unique and can be identified with datasheets or measurements.

**Table 4 sensors-22-06339-t004:** Comparison of TX and RX positions based on their location in the setup.

TX/RX Position	Description
TX Position 1	nRF52840DK board and antenna parallel to left side panel. Located 5.725 cm above inside bottom; 17.8 cm from back-side panel; and 9.25 cm from left side panel. Free space scenario, no metallic object in near field. Emitting radiation of TX is directed towards RX. Direct communication path in range of 60–90° in horizontal plane (see Figure 8) where gain of antenna is close to maximum.
TX Position 2	nRF52840DK board and antenna parallel to left side panel. Located 5.725 cm above the inside bottom; directly adjacent to back-side panel; 9.25 cm from left side panel. main direction of radiation of antenna towards the center of the box. Near field of the antenna influenced by backside panel. This position is used to investigate impact on onboard antenna and external antennas.
TX Position 3	nRF52840DK board and antenna rotated 90° and normal to left side panel. Located 5.7 cm above inside bottom; 26 cm from front panel; 9 cm from left side panel. vertical radiation plane for investigation of differences in radiation efficiency according to Figure 8 and Figure 9. Position 3 to investigate antenna performance in vertical plane. External antennas 20 cm from left side panel; 5 cm above bottom (no impact on near field).
TX Position 4	nRF52840DK board and antenna rotated 180°. Located 5.725 cm above inside bottom; 9.25 cm from the left side panel; directly adjacent to front panel. Focus of investigation horizontal plane. Position 4 used to observe impact of metal in near field of antenna. Antenna directly mounted on the inside of front panel. External antenna position: 20 cm from left side panel; 10 cm above bottom plate.
RX Position	RX position stationary (see Figure 2): envisaged system with star topology and central receiver in dashboard. RX is nRF52840 DK board with onboard monopole antenna. Main direction of radiation of antenna aimed at center of testbed. Here (angle of approx. 90°), radiation efficiency is close to maximum in horizontal plane.

**Table 5 sensors-22-06339-t005:** Complete listing of tested antennas and their data sheets. The main selection criteria were small size (maximum 5 cm in width or height) and the BLE frequency range (2.4 GHz). The size of the antennas is relevant because antennas with a greater size would simply not fit in many of the gaps between components inside a real engine compartment.

Manufacturer	Antenna Designation
Nordic	(1) Monopole nRF52840 DK [47]
Antenova	(1) SR4W030 [52]
Kyocera AVX	(1) 1001932PT-AA10L0100 [53]; (2) 1000418 Wi-Fi Dual Band [54]
Laird	(1) 001-0015 [55]; (2) 001-0034 [56]; (3) CAF94505 [57]; (4) MAF94264 [58]
Molex	(1) 47950-4011 [59]; (2) 146220-0100 [60]; (3) 204281-0100 [61]; (4) 206994-0100 [61]
Pulse Larsen	(1) W3334B0150 [62]
Taoglas	(1) FXP70070053A [63]; (2) FXP74070100A [64]; (3) FXP75070045B [65]; (4) FXP810070100C [66]; (5) WCM010151W [67]
TE Connectivity	(1) 2118059-1 [68]
Yageo	(1) ANTX100P001B24003 [69]; (2) ANTX100P111B24003 [70]; (3) ANTX100P113B24003 [71]

**Table 6 sensors-22-06339-t006:** Comparison of expected and achieved results at positions 1-4 with the vector network analyzer.

Position	Result Description
Position 1	Position reflects undisturbed communication scenario; antenna is radiating in angle with highest gain. Towards receiving antenna. Expected results: antennas work as stated in their data sheets. Achieved results: majority of antennas worked as expected (shown with RSSI and VNA measurements). Antennas with offset had to be matched with adapter implementation (see Section 5.3). Antennas that required matching: AVX 1001932PT-AA10L0100, Taoglas FXP70070053A.
Position 2	Metal in near field of antenna. Antennas aligned for maximum directivity towards receiving antenna. Expected results: some antennas might show frequency shift or will not work when close to metal. Achieved results: many antennas exhibit weakness (as expected) when close to the metal surface (checked with RSSI and VNA measurements). Antennas that performed well: TE Connectivity 2118059-1, Antenova SR4W030, Laird CAF94505, Molex 146220-0100.
Position 3	Position reflects the undisturbed communication scenario. Expected results: reflects the general antenna. Behavior at the defined frequency. Transmission characteristics similar to free space (no near field impact). Achieved results: antennas behave as expected and performance can be derived from the free space scenario. Results are very similar to the ones achieved at position 1.
Position 4	Metal is in or close to the near field of the antenna. Expected results: impact of metal in near field of antennas revealed. Achieved results: some antennas work considerably well, other antennas lose their characteristics and are not suitable for operation anymore. Based on the results: Some antennas can be omitted for positions close to metal in the testbed. However, these antennas might work good for positions with distance to metal. Antennas that performed well: Antenova SR4W030, Yageo ANTXP100P113, Yageo ANTX100P111, TE Connectivity 2118059-1, Taoglas WCM010151W.

**Table 7 sensors-22-06339-t007:** Measured RSSI values and S-parameters $S_{11}$ values inside the BLE frequency band derived from VNA measurements for each scenario: ‘NO’ no object or ‘O’ object is present inside the toolbox; lid ‘LO’ open or ‘LS’ shut (closed). The values between columns representing the same scenario are color-coded according to the assessed quality of the wireless communication—the thresholds for the color codings are shown in the last four rows of the table.

BLE (2.402–2.48 GHz)	RSSI [dBm]: min/avg/max				VNA S-Parameter [dB]: min/avg/max	
Antenna	NO-LO	NO-LS	O-LO	O-LS	NO-LS	O-LS
Antenova SR4W030	−22/−26/−33	**−19**/−20/−21	−21/−25/−31	**−22**/−26/−33	−8.1/−13.2/**−21.9**	−3.4/−7.4/**−12.0**
Antenova SR4W030	90° rotation	−24/−36/−48	90° rotation	−31/−37/−45		
Kyocera AVX1000418	−27/−32/−40	**−19**/−25/−33	−33/−37/−45	**−30**/−44/−65	−9.3/−13.5/**−20.3**	−2.1/−3.5/**−4.1**
Kyocera AVX1001932PT	−22/−32/−43	**−18**/−26/−37	−29/−39/−50	**−26**/−37/−53	−3.9/−7.7/**−14.9**	−1.9/−3.1/**−3.9**
Laird 001-0015	−18/−33/−58	**−20**/−35/−54	−27/−41/−51	**−26**/−38/−50	−5.9/−10.8/**−12.3**	−2.3/−3.6/**−5.1**
Laird 001-0034	−25/−29/−33	**−25**/−27/−30	−36/−41/−45	−38/−43/−50	−6.1/−9.2/**−11.2**	n/a **
Laird CAF94505	−22/−26/−38	**−19**/−25/−36	−32/−38/−45	**−28**/−37/−44	−4.1/−6.5/**−8.4**	−2.1/−2.4/**−3.3**
Laird MAF94264	−22/−36/−46	**−20**/−30/−42	−32/−43/−53	**−32**/−42/−50	−5.2/−7.5/**−10.0**	−1.8/−2.1/**−2.9**
Molex 47950	−33/−36/−47	−28/−32/−41	−38/−44/−61	−34/−43/−49	– *	– *
Molex 146220	−26/−35/−43	**−24**/−35/−51	−35/−44/−47	**−35**/−43/−51	−4.1/−6.9/**−9.0**	−0.8/−1.3/**−2.1**
Molex 204281	−29/−37/−50	−25/−35/−41	−41/−45/−53	−34/−44/−50	n/a **	n/a **
Molex 206994	−18/−32/−43	**−23**/−35/−61	−34/−48/−62	**−28**/−45/−57	−5.0/−8.9/**−11.3**	−4.5/−5.8/**−7.9**
Pulse Larsen W3334B0150	−18/−27/−39	−17/−30/−45	−28/−40/−54	−28/−36/−50	– *	– *
Taoglas FXP70070053A	−17/−21/−27	**−19**/−22/−30	−29/−32/−41	**−26**/−30/−41	−3.2/−4.5/**−6.3**	−1.1/−2.3/**−3.4**
Taoglas FXP74070100A	−20/−33/−46	**−18**/−36/−60	−28/−39/−47	**−30**/−41/−51	−7.2/−9.6/**−13.2**	−2.4/−3.3/**−4.5**
Taoglas FXP75070045B	−26/−32/−45	**−29**/−35/−46	−36/−40/−50	**−31**/−43/−55	−6.8/−9.8/**−14.7**	−2.1/−2.8/**−3.7**
Taoglas FXP810070100C	−29/−33/−42	**−29**/−33/−46	−34/−47/−58	**−35**/−46/−58	−3.8/−5.6/**−7.7**	−1.3/−2.2/**−2.9**
Taoglas WCM010151W	−17/−19/−20	**−17**/−19/−21	−27/−30/−33	**−25**/−32/−43	−6.5/−14.8/**−24.2**	−4.3/−7.4/**−9.2**
Taoglas WCM010151W			90° rotation	−17/−19/−25		
TE Connectivity 2118059	−22/−30/−40	**−18**/−25/−33	−29/−40/−52	**−25**/−39/−53	−7.8/−12.7/**−19.9**	−3.5/−4.7/**−5.9**
Yageo ANTX100P001	−17/−23/−27	**−19**/−25/−31	−29/−33/−36	**−26**/−30/−36	−8.8/−9.9/**−13.2**	−2.3/−3.1/**−4.4**
Yageo ANTX100P111	−17/−20/−28	**−17**/−22/−33	−24/−29/−33	**−24**/−26/−30	−6.2/−12.2/**−23.1**	−4.3/−5.6/**−7.2**
Yageo ANTX100P113	−17/−24/−31	**−17**/−25/−37	−23/−32/−46	**−25**/−30/−34	−8.1/−11.3/**−14.8**	−4.4/−5.2/**−6.3**
GREEN—Best quality		${\mathrm{RSSI}}_{min}\geq -20$		${\mathrm{RSSI}}_{min}\geq -25$	$S_{11,max}\leq -15$	$S_{11,max}\leq -9$
YELLOW		−20>r≥−22		−25>r≥−28	−15<s≤−13	−9<s≤−5
ORANGE		−22>r≥−25		−28>r≥−31	−13<s≤−10	−5<s≤−3.5
RED—Worst quality		$-25>{\mathrm{RSSI}}_{min}$		$-31>{\mathrm{RSSI}}_{min}$	$-10<S_{11,max}$	$-3.5<S_{11,max}$

* Due to a damaged connector, no repeatability measurements were achieved; ** The data acquired with the VNA in these experiments was faulty, the experiment could not be repeated (VNA not available).

**Table 8 sensors-22-06339-t008:** S-parameter $S_{11}$ values inside the BLE frequency band derived either from VNA measurements or EM simulation results. The scenarios are encoded by acronyms: ‘NO’ no object or ‘O’ object is present inside the toolbox; with the ‘LS’ lid shut (closed). The values between columns representing the same scenario are color-coded according to the difference in absolute value of $S_{11,max}$: GREEN $|\Delta S_{11,max}|\leq 3\mathrm{dB}$, YELLOW $3\;\mathrm{dB}<|\Delta S_{11,max}|\leq 7\;\mathrm{dB}$, ORANGE $7\;\mathrm{dB}<|\Delta S_{11,max}|\leq 10\;\mathrm{dB}$ & RED $10\;\mathrm{dB}<|\Delta S_{11,max}|$. Green indicates that there is a small difference between the VNA measurements and the simulations, while red indicates a big difference.

BLE (2.402–2.48 GHz)	VNA S-Parameter [dB]: min/avg/max		Simulation S-Parameter [dB]: min/avg/max	
Antenna	NO-LS	O-LS	NO-LS	O-LS
Antenova SR4W030	−8.1/−13.2/**−21.9**	−3.4/−7.4/**−12.0**	−8.3/−14.8/**−23.2**	−5.1/−8.2/**−9.4**
Kyocera AVX1000418	−9.3/−13.5/**−20.3**	−2.1/−3.5/**−4.1**	−5.8/−8.4/**−12.2**	−2.8/−3.4/**−5.1**
Kyocera AVX1001932PT	−3.9/−7.7/**−14.9**	−1.9/−3.1/**−3.9**	−4.4/−7.8/**−11.5**	−1.3/−1.7/**−2.1**
Laird 001-0015	−5.9/−10.8/**−12.3**	−2.3/−3.6/**−5.1**	−2.4/−4.6/**−6.8**	−1.9/−2.7/**−3.7**
Laird 001-0034	−6.1/−9.2/−11.2	na **	na **	na **
Laird CAF94505	−4.1/−6.5/**−8.4**	−2.1/−2.4/**−3.3**	−8.1/−9.5/**−11.4**	−2.2/−2.5/**−3.6**
Laird MAF94264	−5.2/−7.5/−10.0	−1.8/−2.1/−2.9	***	***
Molex 47950	na *	na *	na *	na *
Molex 146220	−4.1/−6.9/**−9.0**	−0.8/−1.3/**−2.1**	−2.5/−3.7/**−4.5**	−1.2/−1.7/**−2.1**
Molex 204281	na **	na **	na **	na **
Molex 206994	−5.0/−8.9/**−11.3**	−4.5/−5.8/**−7.9**	−4.2/−6.7/**−9.1**	−3.3/−4.7/**−6.1**
Pulse Larsen W3334B0150	na *	na *	na *	na *
Taoglas FXP70070053A	−3.2/−4.5/**−6.3**	−1.1/−2.3/**−3.4**	−6.2/−9.6/**−12.2**	−1.8/−2.6/**−3.2**
Taoglas FXP74070100A	−7.2/−9.6/**−13.2**	−2.4/−3.3/**−4.5**	−13.0/−18.1/**−24.3**	−1.4/−1.9/**−2.2**
Taoglas FXP75070045B	−6.8/−9.8/−14.7	−2.1/−2.8/−3.7	***	***
Taoglas FXP810070100C	−3.8/−5.6/**−7.7**	−1.3/−2.2/**−2.9**	−2.3/−4.2/**−6.5**	−2.1/−2.3/**−2.5**
Taoglas WCM010151W	−6.5/−14.8/**−24.2**	−4.3/−7.4/**−9.2**	−8.3/−12.4/**−14.2**	−5.4/−7.4/**−9.2**
TE Connectivity 2118059	−7.8/−12.7/**−19.9**	−3.5/−4.7/**−5.9**	−8.9/−13.2/**−16.8**	−6.9/−8.2/**−9.8**
Yageo ANTX100P001	−7.8/−12.7/−19.9	−3.5/−4.7/−5.9	***	***
Yageo ANTX100P111	−6.2/−12.2/**−23.1**	−4.3/−5.6/**−7.2**	−12.2/−16.5/**−19.6**	−5.2/−6.5/**−8.3**
Yageo ANTX100P113	−8.1/−11.3/**−14.8**	−4.4/−5.2/**−6.3**	−7.3/−10.5/**−12.9**	−4.1/−4.9/**−6.5**

* Due to a damaged connector, no repeatability measurements were achieved; ** The data acquired with the VNA in these experiments was faulty; the experiment could not be repeated (VNA not available); *** EM antenna model was too inaccurate. The information in the datasheet is either not accurate enough or wrong.

**Table 9 sensors-22-06339-t009:** EM ray-tracing simulation results. Two antennas, namely, the Antenova SR4W030 and the Laird 001-0015 antenna, are compared in the experimental testbed at two different locations. Ray length and ray power for each of the scenarios are displayed with metrics minimum and maximum, as well as the quartiles Q1, Q2, and Q3. In the hits/launched column, we display how many percent of the 1000 launched rays did hit the receiver. The scenarios are encoded by acronyms: ‘NO’ no object or ‘O’ object is present inside the toolbox; with the ‘LS’ lid shut (closed).

Scenario	TX Antenna **	Ray Length (min/Q1/Q2/Q3/max) [m]	Ray Power (min/Q1/Q2/Q3/max) [dBV/m]	Hits/Launched [%]
Pos. 4—NO-LS	Antenova SR4W030	1.36/1.55/1.73/1.93/4.38	−20.47/−16.35/−14.74/−13.11/−0.80	7.0%
Pos. 4—NO-LS	Laird 001-0015	1.37/1.52/1.71/1.90/2.21	−27.44/−23.82/−22.33/−20.97/−12.45	3.2%
Pos. 4—O-LS	Antenova SR4W030	1.74/1.89/2.00/2.11/2.23	−24.89/−21.33/−19.44/−18.04/−17.83	0.8%
Pos. 4—O-LS	Laird 001-0015	n/a *	n/a *	0.0% *
Pos. 3—NO-LS	Antenova SR4W030	1.30/1.55/1.89/2.36/4.42	−18.03/−12.54/−9.47/−7.88/1.98	18.5%
Pos. 3—NO-LS	Laird 001-0015	1.29/1.56/1.94/2.24/4.21	−17.32/−11.37/−8.35/−6.75/2.45	22.8%

* No launched ray reached the receiver before meeting the stopping criteria (too little ray energy); ** The RX Antenna is always the monopole antenna of the nRF52840DK board.

**Table 10 sensors-22-06339-t010:** Antenna ranking orders provided by our recommender algorithm for two scenarios in position 4: ‘NO’ no object or ‘O’ object present inside the toolbox with the ‘LS’ lid shut (closed). The antenna ranking is achieved by sorting the antennas according to the strongest dip on their $S_{11}$ data—i.e., the minimum value—and using only the data of either the VNA-derived or the EM-simulated S-parameters. The ranking shift is color-coded as {Green, Yellow, Orange, Red} by thresholding by ≤{2,4,6}. Green indicates a small shift, while red indicates a big shift. The grayed-out rankings indicate antennas, which for various reasons could not be used for this evaluation—see comments below the table.

			‘NO-LS’: No Object, Lid Shut			‘O-LS’: With Object, Lid Shut		
ID	Antenna	Rank	VNA-Based	EM-Simulation	Shift	VNA-Based	EM-Simulation	Shift
1	Antenova SR4W030	1	17	14	** −4 **	1	18	** −1 **
2	Kyocera AVX1000418	2	20	1	** −1 **	17	1	** −1 **
3	Kyocera AVX1001932PT	3	1	20	** 1 **	11	17	** −3 **
4	Laird 001-0015	4	2	18	** −3 **	20	20	** 0 **
5	Laird 001-0034 **	5	18	17	** 1 **	21	21	** 0 **
6	Laird CAF94505	6	3	21	** −3 **	18	11	** 5 **
7	Laird MAF94264 ***	7	21	2	** 1 **	4	2	** −1 **
8	Molex 47950 *	8	14	13	** 7 **	14	4	** −4 **
9	Molex 146220	9	4	3	** −3 **	2	6	** 2 **
10	Molex 204281 **	10	11	6	** −1 **	3	13	** −3 **
11	Molex 206994	11	9	11	** −3 **	13	16	** 1 **
12	Pulse Larsen W3334B0150 *	12	6	4	** 2 **	6	14	** 3 **
13	Taoglas FXP70070053A	13	16	16	** 0 **	16	3	** 2 **
14	Taoglas FXP74070100A	14	13	9	** 6 **	9	9	** 0 **
15	Taoglas FXP75070045B ***	** 15 **	** 5 **	** 5 **	** 0 **	** 5 **	** 5 **	** 0 **
16	Taoglas FXP810070100C	** 16 **	** 7 **	** 7 **	** 0 **	** 7 **	** 7 **	** 0 **
17	Taoglas WCM010151W	** 17 **	** 8 **	** 8 **	** 0 **	** 8 **	** 8 **	** 0 **
18	TE Connectivity 2118059	** 18 **	** 10 **	** 10 **	** 0 **	** 10 **	** 10 **	** 0 **
19	Yageo ANTX100P001 ***	** 19 **	** 12 **	** 12 **	** 0 **	** 12 **	** 12 **	** 0 **
20	Yageo ANTX100P111	** 20 **	** 15 **	** 15 **	** 0 **	** 15 **	** 15 **	** 0 **
21	Yageo ANTX100P113	** 21 **	** 19 **	** 19 **	** 0 **	** 19 **	** 19 **	** 0 **

* Due to a damaged connector, no repeatability measurements were achieved; ** The data acquired with the VNA in these experiments was faulty; the experiment could not be repeated (VNA not available); *** EM antenna model was too inaccurate. The information in the datasheet is either not accurate enough or wrong.

## Data Availability

Not applicable.

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
