# Peer review of "Towards a Recommender System for In-Vehicle Antenna Placement in Harsh Propagation Environments"

_sensors, 2022, doi:10.3390/s22176339_

Round 1

Reviewer 1 Report

The authors of the expo present a new approach to improve wireless communication in harsh propagation conditions with the goal of achieving higher overall reliability and lifetime of wireless battery-powered in-vehicle sensor systems.
They focus on the investigation of the physical layer and antenna design for a specific hostile environment, i.e., the possibility of interference in the vehicle's engine compartment. They propose the use of electromagnetic (EM) and beamforming simulations as a computationally efficient method to develop an analysis for antenna design and recommendation. They test the functional devices using an experimental test facility - or test environment - that consists of both a physical and an identical test environment setup. A set of antennas is evaluated to identify and verify the behavior of the antennas and characteristics at specified locations in a challenging environment.
Analysis of the experimental measurements and their EM simulation shows that both types of data lead to equivalent antenna results. Then the recommendation for each of the defined locations and experimental conditions is theoretically justifiable. This process of evaluation and validation using the measurements on the experiment is important to validate the antenna selection process.
The results obtained suggest that with properly parameterized and characterized antennas, a number of lengthy and expensive experiments can be replaced by EM simulations on an accurate EM model, which can contribute to significantly speed up the antenna position determination and antenna selection process.
The text of the paper is written based on scientific procedures and rules. It is supplemented by a number of graphical supporting materials. The text does not deal with the design and analysis of individual antenna designs and shapes, but focuses on their application and making verification measurements.
I would recommend to improve (enlarge the font in graphical outputs such as Figure 4, legend of the colour scale, Figure 7 bottom, description of the elevation plots etc., Figure 22 colour legend to the graph, Figure 25 dependency plots and description of the axes, Figure 26 description of the axes.
The text of the paper in a partial way brings novelty in the field.

Reviewer 2 Report

The paper presents a Recommender System for In-Vehicle Antenna Placement in Harsh Propagation Environments. Overall paper is written in easy-to-understand manner. However, following things need clarification for improvement of the paper

1- Good beamforming and reduced side lobe levels are already discussed in research paper 21 and 22. What is need of using again these methods for same purpose.

2- Leaky least mean square algorithm is not elaborated. 

3- How LLMS algorithm is implemented?

4- Normalized Leaky Variable Step Size-LMS algorithm has fast convergence and produces deeper nulls at interference direction as compared to LLMS algorithm hence mention any advantage of using LLMS algorithm over Normalized Leaky Variable Step Size-LMS algorithm.

5- For sidelobe level reduction Sidelobe Sequential Damping (SSD) approach is more useful than array synthesis methods as SSD have superior maximum and average SLL performances and lower processing speeds moreover it has constant SLL convergence profile that is independent on the array size. Mention an advantage of your method to reduce SLL over other methods and techniques.

Reviewer 3 Report

this paper proposes an approach to improve wireless communication in car engine house by using EM simulation, the idea is good and significant . however, some points need to be improved :

1. this paper use a metallic toolbox with rectangle shape as the testbed, however, the appearance shape of real engine house is not rectangle, there should be difference between the rectangle box and the car engine house for EM simulation.

2. in Fig. 11, why the position 1-4 are chosen for the antenna? what is the reasoning behind this choosing?

3. there is no real engine objects put in the testbed to make experiments for comparison of the EM simulation and the real engine compartments test, so the accuracy of the model  can not be verified. 

Round 2

Reviewer 3 Report

the paper has been revised according to the comments.